# Towards Self-Supervised Covariance Estimation in Deep Heteroscedastic Regression

**Megh Shukla**
EPFL

**Aziz Shameem** [†]
IIT Bombay

**Mathieu Salzmann**
SDSC and EPFL

**Alexandre Alahi**
EPFL

**deep-regression.github.io**

## Abstract

Deep heteroscedastic regression models the mean and covariance of the target distribution through neural networks. The challenge arises from heteroscedasticity, which implies that the covariance is sample dependent and is often unknown. Consequently, recent methods learn the covariance through unsupervised frameworks, which unfortunately yield a trade-off between computational complexity and accuracy. While this trade-off could be alleviated through supervision, obtaining labels for the covariance is non-trivial. Here, we study *self-supervised* covariance estimation in deep heteroscedastic regression. We address two questions: (1) How should we supervise the covariance assuming ground truth is available? (2) How can we obtain pseudo-labels in the absence of the ground-truth? We address (1) by analysing two popular measures: the KL Divergence and the 2-Wasserstein distance. Subsequently, we derive an upper bound on the 2-Wasserstein distance between normal distributions with non-commutative covariances that is stable to optimize. We address (2) through a simple neighborhood based heuristic algorithm which results in surprisingly effective pseudo-labels for the covariance. Our experiments over a wide range of synthetic and real datasets demonstrate that the proposed 2-Wasserstein bound coupled with pseudo-label annotations results in a computationally cheaper yet accurate deep heteroscedastic regression.

## 1 Introduction

Deep heteroscedastic regression leverages neural networks as powerful feature extractors to regress the mean and covariance of the target distribution. The target distribution is typically used for downstream tasks such as uncertainty estimation, correlation analysis, sampling, and in bayesian frameworks. The key challenge in deep heteroscedastic regression lies in estimating heteroscedasticity, which implies that the variance of the target is input dependent and variable. Moreover, unlike the mean, the covariance lacks direct supervision and needs to be inferred.

The standard approach without the ground-truth covariance relies on optimizing the negative log-likelihood to jointly learn the mean and covariance (Dorta et al., 2018). However, Skafte et al. (2019); Seitzer et al. (2022) show that in the absence of supervision, incorrect variance predictions lead to sub-optimal convergence. Subsequently, a flurry of recent works addresses this by either modifying the negative log-likelihood (Skafte et al., 2019; Seitzer et al., 2022; Stirn et al., 2023; Immer et al., 2023) or improving the covariance through better parameterization (Shukla et al., 2024). However, the methods introduce a trade-off between computational complexity and accuracy. Moreover, the thematic message underlying these works is that estimating heteroscedasticity is challenging when annotations for the covariance are not available. Therefore, would having annotations for the covariance alleviate this trade-off? To answer this, we explore the use of self-supervision for covariance estimation in deep heteroscedastic regression. We study two questions:

**(Q1) How can we supervise the learning of the covariance assuming annotations are available?**
Since the negative log-likelihood is not formulated for supervising the covariance, we analyse the KL Divergence and the 2-Wasserstein distance to supervise the learning of the mean and covariance.

---

[†] Work done by Aziz Shameem during his internship at EPFL.
Contact: `firstname.lastname@epfl.ch`

Our analysis highlights the utility of the KL Divergence as a regularizer of the covariance. However, we observe that despite supervision, the KL divergence underperforms when compared to the 2-Wasserstein distance, as it shares a susceptibility to residuals similar to that of the negative log-likelihood. Next, we study the 2-Wasserstein distance between normal distributions with non-commutative matrices. We specifically note an optimization challenge (PyTorch, 2024) due to the eigendecomposition involved. Consequently, we extend the formulation for the 2-Wasserstein distance between commutative covariance matrices to the general case of non-commutative matrices, eliminating the need for eigendecomposition. This makes the optimization process stable.

**(Q2) How can we obtain pseudo-labels for the covariance when annotations are not available?** In the absence of priors, we propose a neighborhood-based heuristic algorithm to generate pseudo-labels for the covariance. Specifically, for a given sample, the pseudo-label corresponds to the covariance over the targets of all the samples in the neighborhood of the specified sample. The contribution of the neighboring samples are weighed by their Mahalanobis distance to the specified sample. We show that this simple strategy provides effective self-supervision for covariance estimation.

We perform extensive experiments across a wide range of synthetic and real world settings. Our results show that the proposed self-supervised framework is (1) computationally cheaper and (2) retains accuracy with respect to the state-of-the-art. Finally, our experiments on human pose also highlight the possibility of using self-supervised and unsupervised objectives to further improve optimization. Our code is available at our project page.

## 2 Deep Heteroscedastic Regression

Heteroscedastic regression is the task of modeling the mean and the variance of the target distribution. In contrast to homoscedasticity, heteroscedastic models allow the variance to vary as a function of the input. Deep heteroscedastic regression provides an advantage over non-parametric methods like the Gaussian Processes (Le et al., 2005) because it can model complex features from inputs such as images. This attribute has made it widely applicable in fields like active learning (Houlsby et al., 2011; Gal et al., 2017), uncertainty estimation (Gal & Ghahramani, 2016; Kendall & Gal, 2017; Lakshminarayanan et al., 2017; Russell & Reale, 2021), image reconstruction (Dorta et al., 2018), human pose (Gundavarapu et al., 2019; Nakka & Salzmann, 2023; Tekin et al., 2017), and other vision-based tasks (Lu & Koniusz, 2022; Simpson et al., 2022; Liu et al., 2018; Bertoni et al., 2019).

**Preliminaries.** Our goal is to learn the target distribution $P(Y|X)$ for different $X$, where $X \in \mathbb{R}^m$ is the input and $Y \in \mathbb{R}^n$ is the target variable. While $P(Y|X)$ is unknown, it is assumed to be normally distributed: $P(Y|X) = \mathcal{N}(\mu_Y(X), \Sigma_Y(X))$. Our estimate of the target is $P(\widehat{Y}|X) = \mathcal{N}(\widehat{\mu}_Y(X), \widehat{\Sigma}_Y(X))$, where the mean $\widehat{\mu}_Y(X) = f_\theta(X)$ and covariance $\widehat{\Sigma}_Y(X) = g_\Theta(X)$ are parameterized by neural networks. The standard approach in the literature to learn the target distribution is through minimizing the negative log-likelihood, $-\mathbb{E}_{P(X,Y)}P(\widehat{Y}|X)$. Specifically, the mean and covariance networks are trained to minimize (Nix & Weigend, 1994)

$$\mathcal{L}_{\text{NLL}}(\theta, \Theta) := \mathbb{E}_{P(X,Y)}\left[ \log \left|\widehat{\Sigma}_Y(X)\right| + (Y - \widehat{\mu}_Y(X))^T \ \widehat{\Sigma}_Y^{-1}(X) \ (Y - \widehat{\mu}_Y(X)) \right]. \quad (1)$$

**Challenges.** The lack of supervision for the covariance results in an optimization challenge which is formalized in Skafte et al. (2019); Seitzer et al. (2022). The works observed that an underestimated variance can increase the effective learning rate and disrupt optimization (Skafte et al., 2019), whereas an overestimated variance can decrease the effective learning rate and stop optimization (Seitzer et al., 2022). A number of recent approaches modify the negative log-likelihood to reduce the effect of the predicted covariance during optimization. $\beta$-NLL (Seitzer et al., 2022) scales the negative log-likelihood (Eq. 1) by the predicted variance resulting in the objective: $\mathcal{L}_{\beta-\text{NLL}} = \lfloor \hat{\sigma}(\hat{y})^{2\beta} \rfloor \times \mathcal{L}_{\text{NLL}}$. However, since $\beta$-NLL does not originate from a valid distribution, the optimized values do not estimate the true variance. Stirn et al. (2023) decouples the estimation of the mean and variance by scaling the gradient of the mean estimator with the covariance, thereby eliminating the effect of the predicted covariance on the mean. This leads to conflicting assumptions: the mean estimator assumes that the multivariate residual is uncorrelated while the covariance estimator is expected to identify correlations. Immer et al. (2023) proposed the use of natural parameterization of the univariate normal distribution: $n_1 = \frac{\mu}{\sigma^2}$ and $n_2 = \frac{-1}{2\sigma^2}$ for regression. While principled, the method

assumes univariate targets. TIC-TAC (Shukla et al., 2024), in contrast to previous works, retains the negative log-likelihood and formulates the predicted covariance through the gradient and curvature of the mean. However, the improvement in accuracy comes at the expense of increased computational requirements. A parallel line of works studies the impact of training dynamics in deep heteroscedastic regression. Wong-Toi et al. (2024) provide a theoretical study linking the training of heteroscedastic regression to phase transitions, however the experimental evaluation is limited to univariate outputs. Sluijterman et al. (2024) experimentally show that decoupling the mean and variance networks can lead to improved performance, similar to Stirn et al. (2023). However, while the authors suggested a warm-up schedule, we observed that this may not necessarily improve performance.

An overview of the related works reveals a shared theme: estimating heteroscedasticity is difficult without annotations. Further, existing works trade-off accuracy for lower computational requirements. This trade-off could be mitigated with supervision; however, acquiring labels for the covariance is challenging, which restricts further analysis. Moreover, the negative log-likelihood is not formulated to supervise the covariance, requiring a new approach to supervision. Therefore, we investigate two key aspects of the problem: (1) How can we supervise the covariance when negative log-likelihood is not specifically designed for this task? and (2) How can we generate pseudo-labels for the covariance in the absence of ground truth?

## 3 ANALYSIS

We analyze the KL Divergence and the 2-Wasserstein distance, two widely used metrics for comparing and optimizing distributions. Our analysis focuses on multivariate normal distributions, which follows a common assumption in machine learning that the residuals are normally distributed. We support our analysis by studying Problem 1, which lets us visualize the convergence process of various methods using bivariate normal distributions. We then seek to answer which metric is better suited for deep heteroscedastic regression.

**Problem 1.** (Bivariate Normal Distribution) *We consider the task of learning a bivariate normal distribution. We initialize the target and predicted distributions to have different means. While the predicted distribution is initialized with an identity covariance matrix, the covariance for the target distribution is initialized randomly and such that it exhibits a high degree of correlation ($> 0.5$). Given samples $\boldsymbol{y} \in \mathbb{R}^2$ from the target distribution, the goal is to compare different methods in optimizing the predicted distribution to match the target one.*

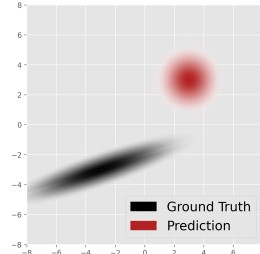

### 3.1 KL DIVERGENCE

The KL Divergence between two continuous distributions $p, q$ is $D_{KL}(p\|q) = \mathbb{E}_p \log[\,p(x)/q(x)\,]$. For two multivariate normal distributions (Zhang et al., 2024; Soch, 2020) we have

$$D_{\text{KL}}(p \parallel q) = \frac{1}{2}\left[\text{Tr}(\Sigma_q^{-1}\Sigma_p) + (\mu_q - \mu_p)^\top \Sigma_q^{-1}(\mu_q - \mu_p) - k + \ln\left(\frac{\det \Sigma_q}{\det \Sigma_p}\right)\right], \quad (2)$$

Since the KL Divergence is asymmetric, we often let $p$ and $q$ be the target and predicted distributions respectively. This definition also gives rise to popular alternatives such as the negative log-likelihood. While the KL Divergence has been well explored (Goodfellow et al., 2016; Arjovsky et al., 2017) from a statistical viewpoint, we ask: how can the KL Divergence be formulated for deep heteroscedastic regression? Answering this is necessary since the KL Divergence is defined in terms of the means and covariances, which are unknown for the target distribution.

**Formulation.** A logical approach would be to replace each label with a distribution. Specifically, for a sample $\boldsymbol{x}_i, \boldsymbol{y}_i$ from the dataset, the pseudo target distribution can be set to $\mathcal{N}(\boldsymbol{y}_i, \Sigma_Y^{(\text{prior})}(X))$. However, this approach requires calibrating the KL Divergence since the optimal value for the covariance is not the same as the prior. We demonstrate this through a simple setting in Lemma 1.

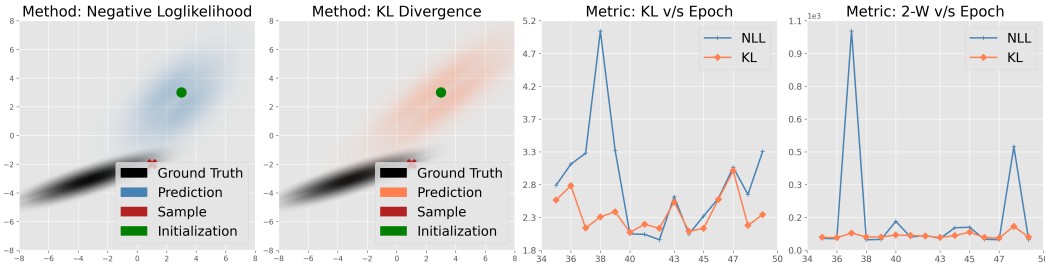

Figure 1: *Sub-optimal convergence due to residuals (Section: 3.1)*. In addition to feature granularity (Seitzer et al., 2022), subpar convergence may occur due to the sensitvity of the negative log-likelihood and the KL-Divergence to residuals in the mini-batch. While we show that the KL-Divergence can act as a regularizer over the learnt covariance, the gradients for both the methods are dominated by the residual term, slowing down convergence.

**Lemma 1** (Calibration). *Let $\mathbf{S} = \{\mathbf{x}, \mathbf{y_i}\}_{i=1}^{N}$ be a set of samples drawn from the unknown target $P(Y|X) = \mathcal{N}(\mu_Y(X), \Sigma_Y(X))$ for a given $\boldsymbol{x}$. We write each label $\boldsymbol{y}_i$ as a distribution $\mathcal{N}(\boldsymbol{y}_i, \Sigma_Y^{(prior)}(X))$. Then, the optimal solution using the KL Divergence for the predicted covariance over the set $\mathbf{S}$ is $\widehat{\Sigma}_Y(X) \approx \Sigma_Y(X) + \Sigma_Y^{(prior)}(X)$. Consequently, if the target covariance is known and set as the prior, we have $\widehat{\Sigma}_Y(X) \approx 2\,\Sigma_Y(X)$.*

We refer the reader to the appendix, section (A.2) for the proof.

**Discussion.** The optimal solution for $\widehat{\Sigma}_Y(X)$ is twice the target covariance. This can be addressed by a simple calibration of the KL Divergence (Eq. 2);

$$\frac{1}{2}\left[\frac{\mathrm{Tr}(\widehat{\Sigma}_Y^{-1}\Sigma_Y^{(prior)}) + (\widehat{\mu}_Y - \boldsymbol{y})^{\top}\widehat{\Sigma}_Y^{-1}(\widehat{\mu}_Y - \boldsymbol{y})}{2} - k + \ln\left(\frac{\det\widehat{\Sigma}_Y}{\det\Sigma_Y^{(prior)}}\right)\right].$$

In the general scenario where the true covariance is not known *a priori*, the optimal solution is the average of the prior and the sample covariance. This introduces a notion of regularization on the predicted covariance. Finally, the predicted covariance is also robust to outliers.

**Impact of residuals.** In general, the solution in Lemma 1 is reached *only* when the mean estimator has converged to the true mean and when we observe multiple targets $\boldsymbol{y}_i$ for the same observation $\boldsymbol{x}$. This may not hold true in practical settings because: (1) samples in a batch are *i.i.d*, implying that the same observation $\boldsymbol{x}$ is unlikely to be repeated, and (2) the mean estimator may not have converged.

In practice, for each sample in the batch, we take a noisy gradient step towards $\widehat{\Sigma}_Y(X) = \Sigma_Y^{(prior)}(X) + (\widehat{\mu}_Y(X) - \boldsymbol{y})(\widehat{\mu}_Y(X) - \boldsymbol{y})^T$ (appendix/eq.13). If the residual term $(\widehat{\mu}_Y(X) - \boldsymbol{y})$ is large, the gradient step due to the residual dominates over the prior covariance, moving us closer to $\widehat{\Sigma}_Y(X) \approx (\widehat{\mu}_Y(X) - \boldsymbol{y})(\widehat{\mu}_Y(X) - \boldsymbol{y})^T$. This residual matrix can be interpreted as a 'covariance' matrix aligned along the line segment joining $\boldsymbol{y}$ and $\widehat{\mu}_Y(X)$ (Eq. 3; OLS estimate of the slope, Soch (2021)). However, this residual matrix desensitizes the mean estimator to move along $\boldsymbol{y}$ and $\widehat{\mu}_Y(X)$, slowing down optimization. This is because the 'variance' induced by the residual matrix is large along $\boldsymbol{y}$ and $\widehat{\mu}_Y(X)$, and the gradient of the mean estimator is proportional to the inverse of the covariance (appendix/eq.11). This is pictorially depicted in appendix/Fig. 7.

We study this phenomenon through Problem 1 in Fig. 1. We observe that after a few iterations, the predicted covariance is aligned along the means of the target and predicted distribution. This observation is a result of the residual term appearing in the optimal solution. Moreover, while the KL Divergence incorporates our prior knowledge of the covariance, the prior term is dwarfed in magnitude when compared to the residual term. We also note increased optimization instability at higher learning rates (appendix/fig. 8). While the KL Divergence leverages the prior covariance as a regularizer, it shares drawbacks pertaining to the residual with the negative log-likelihood, motivating our analysis of the 2-Wasserstein distance.

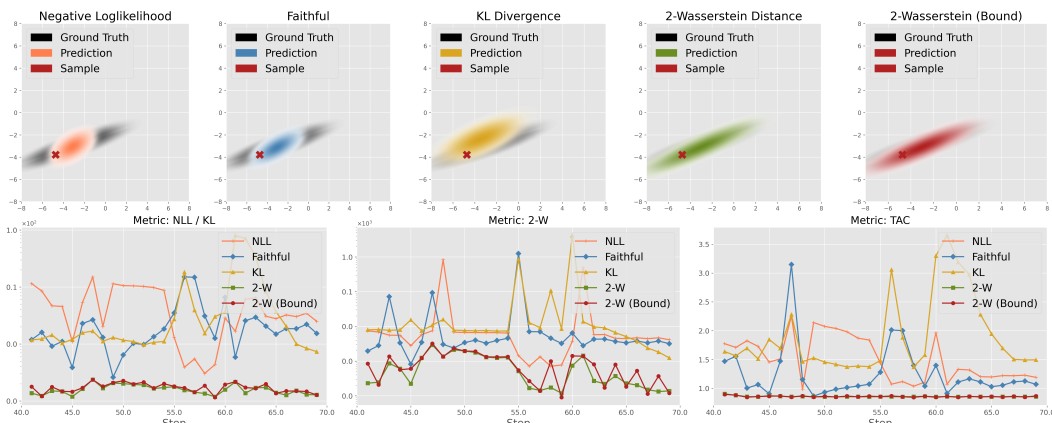

Figure 2: *Visualizing convergence in bivariate regression (Section: 3.2).* We observe that the KL-Divergence and likelihood based methods: vanilla negative log-likelihood and Faithful (Stirn et al., 2023) result in unstable convergence due to the sensitivity of the methods to the residuals. In comparison, the 2-Wasserstein based methods are more stable and accurate. This observation can also be replicated when the predicted mean is initialized at the same location as the true mean, shown in appendix/Fig.9 (b) *(Note: metrics NLL / KL and 2-W are plotted in log-scale)*

## 3.2 2-WASSERSTEIN DISTANCE

The Wasserstein distance is a metric for quantifying the distance between two probability distributions. It defines the minimum "cost" required to morph one distribution into another. The 2-Wasserstein distance measures the cost in proportion to the squared Euclidean distance. It is widely used in optimal transport theory and generative modeling (Arjovsky et al., 2017; Li et al., 2024), as it captures both the shape and spread of distributions while penalizing long-distance transport more heavily. Let $\mathcal{N}_1(\mu_1, \Sigma_1)$, $\mathcal{N}_2(\mu_2, \Sigma_2)$ be two multivariate normal distributions. The 2-Wasserstein distance between them is given by

$$||\mu_1 - \mu_2||^2 + \text{Tr}\big[\Sigma_1 + \Sigma_2 - 2(\Sigma_2^{1/2}\Sigma_1\Sigma_2^{1/2})^{1/2}\big] \ . \tag{3}$$

This formulation, however, requires computing the root of a matrix, which typically involves eigendecomposition. Unfortunately, the eigendecomposition in popular deep learning frameworks can potentially lead to unstable gradients (PyTorch, 2024). If $\Sigma_1$ and $\Sigma_2$ are commutative (implying $\Sigma_1\Sigma_2 = \Sigma_2\Sigma_1$), then the 2-Wasserstein distance is reduced to $\mathcal{W}_2(\mathcal{N}_1, \mathcal{N}_2) = ||\mu_1 - \mu_2||^2 + ||\Sigma_1^{1/2} - \Sigma_2^{1/2}||_F^2$. This formulation allows us to directly predict the square roots, thereby avoiding the eigendecomposition. However, for two covariance matrices to be commutative, they need to share the same eigenbasis, implying that the matrices differ only in the variance of the individual random variables. Fortunately, Theorem 1 allows us to expand this formulation to non-commutative covariance matrices by linking it to an upper bound on the true 2-Wasserstein distance.

> **Theorem 1** (2-Wasserstein bound for non-commutative covariances). *Let $\mathcal{N}_1(\mu_1, \Sigma_1)$, $\mathcal{N}_2(\mu_2, \Sigma_2)$ be two multivariate normal distributions, where $\Sigma_1$ and $\Sigma_2$ are non-commutative matrices. Then, the 2-Wasserstein distance between the two distributions has an upper bound of*
>
> $$\mathcal{W}_2(\mathcal{N}_1, \mathcal{N}_2) \leq ||\mu_1 - \mu_2||_2^2 + ||\Sigma_1^{1/2} - \Sigma_2^{1/2}||_F^2 \ ,$$
>
> *where $||(.)||_F$ represents the Frobenius norm of a matrix.*

We refer the reader to the appendix, section (A.1) for the proof.

**Significance.** Deriving this bound allows us to extend the simplification for the 2-Wasserstein distance between two commutative covariance matrices to the more general scenario of non-commutative matrices. The simplification allows us to directly supervise the covariance without the use of eigendecomposition, making optimization inherently more stable.

**Algorithm 1:** *Covariance Pseudo-Label*

**Input:** $x \in \mathbb{R}^M$: Given observation
**Input:** $\mathcal{X}$: All observations; $\mathcal{Y}$: All targets

**Output:** $\widetilde{\Sigma}_Y(X)$: Covariance pseudo-label

```
// Mahalanobis distance
```
$\Sigma = \mathrm{Cov}(\mathcal{X})$

```
// d(x, X, Σ).shape = #samples
```
$d_M(x, \mathcal{X}, \Sigma) = (\mathcal{X} - x)\Sigma^{-1}(\mathcal{X} - x)^T$

```
// k = Nearest neighbours
```
dist, idx = `bottom-k` $(d_M(x, \mathcal{X}, \Sigma), \text{k})$

```
// 'Probabilistic' interpret
```
dist = `softmax` (dist)
```
// y's for nearest neighbours
```
y_nbr = $\mathcal{Y}$ [idx]

```
// 'Expected' mean, covariance
```
$\tilde{\mu}_y$ = (dist * mean).`sum(dim=0)`
$\widetilde{\Sigma}_Y(X)$ = dist * (y_nbr $- \tilde{\mu}_y$)(y_nbr $- \tilde{\mu}_y)^T$

```
// Pseudo-label for given x, y
```
**return** $\widetilde{\Sigma}_Y(x)$

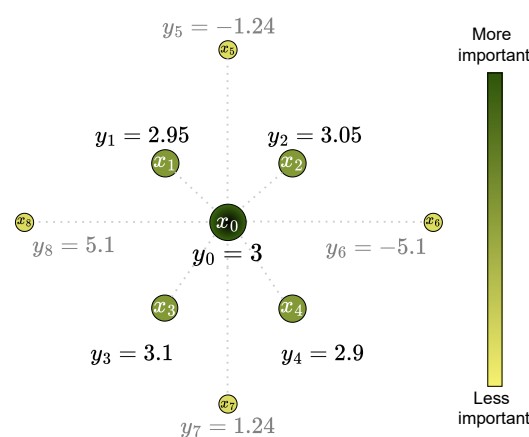

Figure 3: *Pseudo-Label (Section 3.3)* Given $x_0$, its pseudo-label is the variance in the targets $y$ corresponding to samples which are the nearest neighbors of $x_0$. Samples closer to $x_0$ are given more importance than samples further away.

We return to Problem 1 and visually compare the 2-Wasserstein distance with variants of the log-likelihood (vanilla negative log-likelihood, Faithful (Stirn et al., 2023)) and KL-Divergence. In Fig. 2, we observe that in comparison to the likelihood based methods, the 2-Wasserstein is significantly more stable since the covariance does not depend on the residual and the convergence of the mean estimator. We also study the impact of warm-up (Sluijterman et al., 2024), where the mean estimator is allowed to converge before training the covariance estimator. To do so, we directly initialize the mean of the predicted distribution to the target mean, and the covariance to identity. However, our results in appendix/Fig.9 (b) (appendix) show that these methods are still susceptible to instability due to residuals. In contrast, the learning of the covariance in 2-Wasserstein does not depend upon the residual, leading to stable convergence.

### 3.3 GENERATING PSEUDO-LABELS FOR THE COVARIANCE

In the absence of labels for the covariance, existing approaches rely on the residual of the mean estimator as a signal to optimize the covariance. However, optimizing in this manner trades-off accuracy with computational complexity. While having labels would allow us to directly optimize the covariance estimator, obtaining annotations for the covariance is non-trivial. Therefore, we take a step in this direction and explore the possibility of self-supervision for the covariance. To this end, we propose a simple heuristic, which when combined with the 2-Wasserstein distance, is surprisingly effective in supervising the covariance.

**Intuition.** The neighborhood of a sample has been widely used in uncertainty quantification (Van Amersfoort et al., 2020; Skafte et al., 2019) and kernel methods (Hofmann et al., 2008). The key idea is to infer properties of a sample based on its neighborhood. TIC (Shukla et al., 2024) learns the covariance through a learnt $\epsilon-$neighborhood of the input, $\mathrm{Cov}(\hat{Y}|X + \epsilon)$. We extend upon this idea to obtain pseudo-labels for the covariance. Specifically, we use two concepts:

1. The target $y$ has a high (co-)variance if it exhibits large variations in a small vicinity of $x$.
2. The closer $x_j$ is to $x_i$, the likelier it is that $y_j$ is a potential label for $x_i$.

We quantify these concepts through the use of (a) the Mahalanobis distance, to measure the degree of closeness between samples $x_i$ and $x_j$; and (b) a probabilistic interpretation of this distance to weight different targets $y_j$ as being a potential label for $x_i$.

The Mahalanobis distance between two points $\mathbf{u}, \mathbf{v}$ with respect to a covariance matrix $\Sigma$ is

$$d_M(\mathbf{u}, \mathbf{v}; \boldsymbol{\Sigma}) := \sqrt{(\mathbf{u} - \mathbf{v})^{\mathbf{T}}\boldsymbol{\Sigma}^{-1}(\mathbf{u} - \mathbf{v})}. \tag{4}$$

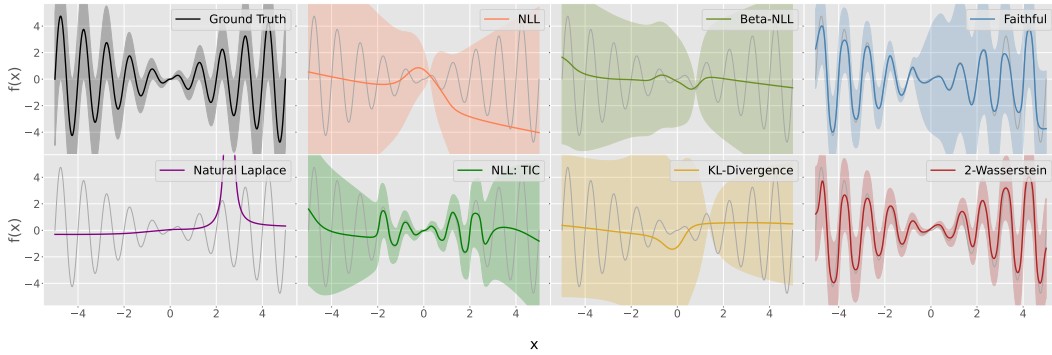

Figure 4: We sample from the ground truth sinusoidal $y = |x| \sin(2\pi x)$ with $\sigma(x) = |x|$ and train our networks using different objectives. The 2-Wasserstein distance trained using pseudo-labels is able to converge to the accurate mean and variance faster since it does not depend upon residuals or convergence of the mean estimator to learn the variance.

Unlike the Euclidean distance, the Mahalanobis distance accounts for the spread of the samples. This is crucial since not only does the distance scale according to the alignment of $\mathbf{u}, \mathbf{v}$ *w.r.t.* the covariance, but it also scales based on the spatial extent of the samples.

Therefore, we define $\Sigma = \text{Cov}(X)$ to quantify the alignment and scale of all the samples $X$. We compute the pairwise Mahalanobis distance between the given sample and all other samples, choosing the *top-k* nearest neighbors and their associated distances. This also includes the given sample itself. Next, we compute the softmax over these distances, giving them a probabilistic interpretation: the closer the sample, the higher the likeliness of it being the true mean. The pseudo-label covariance uses this probabilistic interpretation to compute the expected mean and covariance over the neighboring targets. A concise description of these steps is available in Algorithm 1.

**Self-Supervised Training**. We use the pseudo-labels $\widetilde{\Sigma}_Y(\boldsymbol{x})$ in conjunction with the input $\boldsymbol{x}$ and target $\boldsymbol{y}$ to train the mean and covariance estimators simultaneously using the 2-Wasserstein bound.

## 4 EXPERIMENTS

How effective is self-supervision in deep heteroscedastic regression? We study this question through a series of synthetic and real world datasets for regression. We use the same setup as Shukla et al. (2024) which experiments on univariate sinusoidals, synthetic multivariate data, UCI Machine Learning repository (Markelle Kelly) and 2D human pose estimation (Andriluka et al., 2014; Johnson & Everingham, 2010; 2011). We provide a detailed description of the experimental setup and implementation details in the appendix (B). For all our experiments, we set the nearest neighbors hyperparameter in the pseudo-label algorithm to ten times the dimensionality of the target. We compare our approach with popular and state-of-the-art methods such as the vanilla negative log-likelihood, $\beta-$NLL (Seitzer et al., 2022), Faithful heteroscedastic regression (Stirn et al., 2023), Empirical-Bayes (Immer et al., 2023) and the Taylor Induced Covariance parameterization (NLL:TIC) (Shukla et al., 2024). In addition to the mean square error and the negative log-likelihood, we use the KL-Divergence and the 2-Wasserstein distance as metrics when the ground truth covariance is known. We also use the Task Agnostic Correlations (TAC) metric introduced in (Shukla et al., 2024) to evaluate the covariance through its learnt correlations. Finally, we also report the additional memory consumed and the time required for each method to run for different experiments.

### 4.1 SYNTHETIC DATA

**Univariate.** We use samples from different varying amplitude sinusoidals to compare the methods. In Fig. 4, we observe that faithful heteroscedastic regression overestimates the covariance because of the lack of synergy between the mean and variance estimator. While the mean estimator assumes homoscedastic unit variance, the variance estimator models heteroscedasticity. Although the TIC formulation stabilizes convergence, unfortunately, convergence itself is slow. The KL Divergence

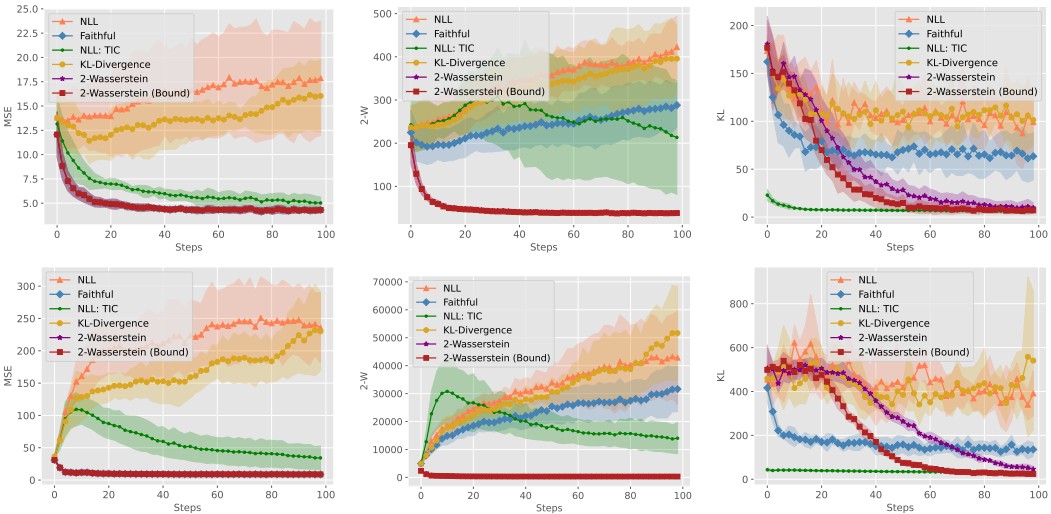

Figure 5: *(Multivariate: Metrics.)* We simulate multivariate data with heteroscedastic covariance of increasing dimensionality (top row: 8, bottom row: 24). We observe that modeling heteroscedasticity is challenging without annotations, with some popular approaches diverging away from the true distribution. Our results highlight the potential of self-supervision for improved convergence.

Table 1: *(Multivariate: Computational Costs).* While TIC is able to accurately model the covariance in comparison to other likelihood based approaches, it has a significantly increased computational cost. The 2-Wasserstein (bound) has a significantly lower cost without sacrificing accuracy.

(a) Compute time (in milliseconds)

| Dimensions → | 4 | 8 | 12 | 16 | 20 | 24 | 28 | 32 |
|---|---|---|---|---|---|---|---|---|
| Beta-NLL, Diagonal | 2.88 | 3.15 | 2.17 | 2.06 | 1.83 | 1.74 | 2.00 | 2.04 |
| Faithful, NLL | 4.56 | 4.74 | 3.94 | 3.69 | 3.76 | 3.66 | 4.08 | 4.85 |
| NLL: TIC | 56.60 | 56.81 | 59.28 | 95.54 | 197.58 | 448.58 | 943.79 | 1961.08 |
| KL-Divergence | 4.79 | 5.06 | 4.05 | 4.05 | 4.10 | 3.94 | 4.81 | 5.24 |
| 2-Wasserstein | 5.10 | 5.43 | 4.47 | 4.38 | 4.31 | 4.14 | 4.88 | 5.20 |
| 2-Wasserstein (Bound) | 4.59 | 4.79 | 3.72 | 3.73 | 3.64 | 3.56 | 3.91 | 4.72 |

(b) Compute memory (in megabytes)

| Dimensions → | 4 | 8 | 12 | 16 | 20 | 24 | 28 | 32 |
|---|---|---|---|---|---|---|---|---|
| NLL: TIC | 11.68 | 120.84 | 523.22 | 1543.77 | 3625.51 | 7333.84 | 13313.31 | 22398.00 |
| 2-Wasserstein (Bound) 
 +6 other methods | 3.45 | 8.24 | 17.10 | 29.51 | 54.51 | 111.73 | 201.51 | 339.55 |

and vanilla negative log-likelihood suffer from large residuals which prevents further optimisation. In comparison, the 2-Wasserstein distance combined with the pseudo-labels is able to converge faster while being accurate. An additional study comparing the methods on different variations of the sinusoidal is presented in the appendix (Fig: 10).

**Multivariate**. Unlike in our real world experiments, synthetic datasets allow us to define the ground truth covariance to evaluate different approaches. We use the same setup as previous work, which defines the multivariate target $\mathcal{N}(\mu_{Y|X}, \Sigma_{Y|X} + \Sigma_{Z|X})$ as a function of the input with heteroscedastic variance. $X$ and $Y$ are jointly distributed following the normal distribution, with $Z$ being a variable conditionally independent of $Y$ and a function of $X$. In addition to evaluating different methods through optimization metrics, we also compare them through their computational requirements (memory and time). We vary the dimensionality of our targets ranging from 4 to 32, and report our results in Fig. 5, 11 (appendix), and Table 1. We observe that while TIC facilitated improved

Table 2: *UCI Regression*. The 2-Wasserstein distance using pseudo-labels for supervision accurately estimates the mean and covariance while having low compute requirements. In contrast, the negative log-likelihood and KL Divergence sub-optimally converge due to large residual errors. While methods such as Faithful encourage convergence by using the mean squared error, the mean and covariance inconsistently model the residual leading to sub-optimal covariance estimates. While TIC accurately models the covariance, it has high computational costs and lags in mean estimation.

(a) Mean Square Error (MSE)

| Method | Abalone | Air | Appliances | Concrete | Electrical | Energy | Gas | Naval | Parkinson | Power | Red Wine | White Wine |
|---|---|---|---|---|---|---|---|---|---|---|---|---|
| NLL | 3.74 | 17.92 | 53.49 | 4.57 | 9.28 | 4.20 | 10.98 | 10.34 | 54.51 | 9.09 | 8.94 | 9.40 |
| KL-Divergence | 1.90 | 14.70 | 90.90 | 3.84 | 15.57 | 4.20 | 10.16 | 12.39 | 59.39 | 9.97 | 7.26 | 8.17 |
| Beta-NLL | 0.35 | 1.58 | 3.69 | 2.02 | 3.62 | 1.87 | 1.50 | 0.72 | 8.11 | 3.06 | 2.15 | 3.43 |
| NLL: Diagonal | 1.32 | 8.90 | 37.91 | 4.28 | 6.58 | 3.99 | 5.73 | 9.60 | 27.35 | 6.52 | 5.75 | 6.01 |
| Faithful | **0.16** | **0.33** | **0.20** | **0.72** | **0.89** | **0.41** | **0.45** | **0.06** | **0.29** | **0.61** | **0.70** | **0.78** |
| NLL: TIC | 0.21 | 0.82 | 4.45 | 0.96 | **0.91** | 0.61 | 0.67 | 1.36 | 8.89 | **0.66** | 0.97 | 0.92 |
| **2-W (Bound)** | **0.16** | **0.34** | **0.20** | **0.72** | **0.90** | **0.41** | **0.45** | **0.07** | **0.30** | **0.61** | **0.71** | **0.79** |

(b) Negative Log-Likelihood (NLL)

| Method | Abalone | Air | Appliances | Concrete | Electrical | Energy | Gas | Naval | Parkinson | Power | Red Wine | White Wine |
|---|---|---|---|---|---|---|---|---|---|---|---|---|
| NLL | 35.89 | 56.98 | 245.99 | 28.50 | 63.15 | 29.85 | 41.03 | 38.18 | 262.59 | 49.37 | 46.58 | 58.06 |
| KL-Divergence | 18.27 | 83.18 | 413.96 | 38.23 | 73.87 | 28.50 | 34.38 | 49.24 | 257.36 | 45.49 | 61.96 | 48.66 |
| Beta-NLL | 9.80 | 29.38 | 60.30 | 20.45 | 35.44 | 20.15 | 20.26 | 20.81 | 59.98 | 27.64 | 34.05 | 30.95 |
| NLL: Diagonal | 18.61 | 80.86 | 369.67 | 46.82 | 65.73 | 36.09 | 47.06 | 77.47 | 238.71 | 51.60 | 77.98 | 67.21 |
| Faithful | 11.86 | 33.31 | 65.15 | 17.42 | 34.73 | 19.41 | 22.47 | 27.70 | 57.04 | 24.08 | 24.34 | 26.01 |
| NLL: TIC | **4.71** | 16.46 | 30.41 | 11.36 | **14.97** | 12.06 | **9.96** | 14.99 | 42.52 | **9.31** | 14.66 | **12.33** |
| **2-W (Bound)** | 6.32 | **13.58** | **22.72** | **8.96** | 15.57 | **8.85** | 10.49 | **11.44** | **21.48** | 11.31 | **11.65** | 12.12 |

(c) Compute time (in milliseconds)

| Method | Abalone | Air | Appliances | Concrete | Electrical | Energy | Gas | Naval | Parkinson | Power | Red Wine | White Wine |
|---|---|---|---|---|---|---|---|---|---|---|---|---|
| NLL | 5.28 | 5.45 | 5.85 | 5.70 | 7.53 | 5.84 | 5.23 | 6.44 | 7.31 | 5.53 | 5.95 | 5.88 |
| Beta-NLL | 4.71 | 4.58 | 4.98 | 4.35 | 5.20 | 4.41 | 4.62 | 5.81 | 6.70 | 4.81 | 6.83 | 4.41 |
| Faithful | 5.28 | 5.35 | 5.73 | 5.62 | 6.02 | 5.03 | 5.03 | 7.02 | 6.47 | 5.81 | 7.40 | 5.05 |
| NLL: Diagonal | 4.50 | 4.49 | 4.84 | 4.67 | 5.13 | 4.32 | 4.38 | 5.84 | 4.98 | 4.61 | 5.77 | 4.36 |
| NLL: TIC | 45.61 | 53.22 | 68.55 | 47.46 | 49.37 | 47.57 | 45.09 | 56.04 | 59.25 | 49.74 | 65.25 | 45.23 |
| KL-Divergence | 5.30 | 6.65 | 6.08 | 5.47 | 8.09 | 5.16 | 5.14 | 6.85 | 9.15 | 5.36 | 6.93 | 5.23 |
| 2-W (Bound) | 4.51 | 5.83 | 5.17 | 4.51 | 5.36 | 4.50 | 4.38 | 5.23 | 7.16 | 4.51 | 5.28 | 4.48 |

(d) Compute memory (in megabytes)

| Method | Abalone | Air | Appliances | Concrete | Electrical | Energy | Gas | Naval | Parkinson | Power | Red Wine | White Wine |
|---|---|---|---|---|---|---|---|---|---|---|---|---|
| NLL: TIC | 11.22 | 90.20 | 820.85 | 1.09 | 71.35 | 24.13 | 33.99 | 108.59 | 637.74 | 41.57 | 40.94 | 41.57 |
| 2-W (Bound) +6 other methods | 3.10 | 9.02 | 25.30 | 1.09 | 7.75 | 4.63 | 5.56 | 9.02 | 23.05 | 5.56 | 5.56 | 5.56 |

covariance estimation, this resulted in slower convergence of the mean estimator. This trend is evident as the dimensionality increases. Moreover, TIC requires significantly more computational resources. In contrast, the 2-Wasserstein bound is significantly cheaper to compute while maintaining accurate convergence of both, the mean and covariance estimator.

## 4.2 REAL DATASETS

**UCI Regression.** We evaluate mean and covariance predictions by performing the same study as Shukla et al. (2024) on regression datasets from the UCI Machine Learning repository. We standardize each dataset to have zero mean and unit variance. We randomly choose 25% of the features as observations and the remaining 75% as the targets, adding considerable heteroscedasticity in the data. We conduct five trials and report the mean, highlighting top-performing methods which are statistically indistinguishable. We evaluate different methods not only using performance on various metrics (Table 2, appendix/Table 3) but also through computational costs. While TIC outperforms other likelihood based baselines significantly in our TAC and NLL evaluation, this comes at the cost of significantly higher computational requirements. Although compute efficient methods such as Faithful leverage the mean squared error to accurately converge to the mean, it does not accurately converge on the optimal covariance. In contrast, the 2-Wasserstein bound accurately converges in both, the mean and covariance without additional computation overhead. Moreover, the vanilla 2-Wasserstein formulation exhibited significant training instabilities on certain datasets such as *superconductivity*, motivating the use of the proposed bound which is stable to train.

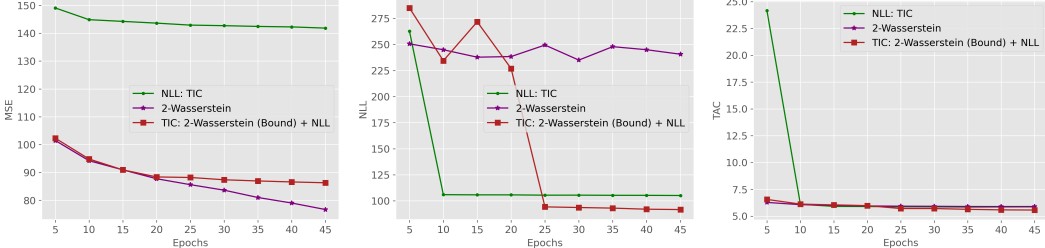

Figure 6: *(Human Pose: Improving state-of-the-art heteroscedastic pose estimation)* We explore a hybrid training strategy by combining the 2-Wasserstein bound with the negative log-likelihood. We train ViTPose for the first 20 epochs using the bound and then switch to negative log-likelihood. We observe that the hybrid approach retains best of both the worlds: improved mean and covariance estimates, as measured by the mean square error and the log-likelihood. *(Different learning rates are explored in Fig. 13)*

We also study the impact of warmup (Sluijterman et al., 2024) in the training process, where we train the mean estimator for half the number of epochs, and jointly train the mean and covariance for the remaining half. We share our results in appendix/Fig. 12. Noticeably, the training diverges due to the effect of residuals coupled with incorrect covariance estimates, effectively nullifying the use of warm-up.

**Human Pose Estimation**. We perform experiments on 2D human pose estimation using the same setup as previous work. We use the ViTPose (Xu et al., 2022) architecture as our base model, which is a popular vision transformer model for human pose estimation. Since we are introducing the covariance in the training process, this also requires us to modify the covariance in response to image and keypoint augmentations. Popular augmentations use affine transformations, which linearly transform the keypoints. Let $\tilde{Y} = RY$ represent the transformed keypoints using the matrix $R$. The new covariance underlying $\tilde{Y}$ is $\tilde{\Sigma}_Y(X) = R\,\widehat{\Sigma}_Y(X)\,R^T$.

We highlight this experiment not as one where self-supervision is computationally efficient, but one which motivates the use of hybrid training to further improve the state-of-the-art. Our preliminary experiments using self-supervision fell short of matching Shukla et al. (2024), which we believe is because the pseudo-labels, computed based on a low-dimensional representation of the input images, may not necessarily be accurate. However, we identify an alternate paradigm: pre-training using self-supervision and then switching to NLL:TIC helped retain the advantages across both, the 2-Wasserstein bound and NLL:TIC.

We show our results in Fig. 6 and Fig. 13. We use the TIC parameterization and train our models using the self-supervision for the first 20 epochs. After this, we switch to the negative log-likelihood which provides more freedom to explore the optimal covariance. Our results show that the hybrid approach outperforms its individual components and retains both: a low mean square error and low negative log-likelihood.

## 5   CONCLUSION

We study deep heteroscedastic regression, noting the optimization challenges present due to the lack of annotations for the covariance. Therefore, we study methods for self-supervision which requires us to define (1) a framework for supervision, and (2) a method to obtain pseudo-labels for the covariance. We critically study the KL-Divergence, highlighting the need for calibration and noting its susceptibility to residuals. Next, we study the 2-Wasserstein distance, proposing a bound on the latter that is stable to optimize. Finally, we propose a simple neighborhood based heuristic which is effective in providing pseudo-labels for the covariance. Our experiments show that, unlike existing approaches, the use of the 2-Wasserstein bound and pseudo-labels yields accurate mean and covariance estimation while remaining computationally inexpensive. Our experiments on human pose show the potential for a hybrid approach, where combining the 2-Wasserstein and NLL frameworks enables superior performance compared to using either method alone.

## REPRODUCIBILITY STATEMENT

We make our code available on `https://deep-regression.github.io`. The code comes complete with a docker image and documentation for reproducibility. We have taken sufficient care to perform multiple trials and report the mean and standard deviation.

## ACKNOWLEDGEMENT

We thank the reviewers for their valuable comments, questions as well as participation in the rebuttal period. We also thank Reyhaneh Hosseininejad for her help in preparing the paper and for having insighful discussions. Finally, we are grateful to RCP, EPFL for their support in compute.

This research is funded by the Swiss National Science Foundation (SNSF) through the project *Narratives from the Long Tail: Transforming Access to Audiovisual Archives* (Grant: CRSII5_198632). The project description is available on: `https://www.futurecinema.live/project/`

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

## A APPENDIX

### A.1 PROOF OF THEOREM 1

*Proof.* We begin with the definition of the 2-Wasserstein distance (Definition 3). First, we focus on rewriting $\Sigma_1 + \Sigma_2$ by adding (and subtracting) new terms to get

$$\Sigma_1 + \Sigma_2 - \Sigma_1^{1/2}\Sigma_2^{1/2} - \Sigma_2^{1/2}\Sigma_1^{1/2} + \Sigma_1^{1/2}\Sigma_2^{1/2} + \Sigma_2^{1/2}\Sigma_1^{1/2}$$

The advantage of introducing new terms is to write $\Sigma_1 + \Sigma_2$ as

$$\Sigma_1 + \Sigma_2 = (\Sigma_1^{1/2} - \Sigma_2^{1/2})(\Sigma_1^{1/2} - \Sigma_2^{1/2})^T + \Sigma_1^{1/2}\Sigma_2^{1/2} + \Sigma_2^{1/2}\Sigma_1^{1/2}\,.$$

Substituting this in the definition of the 2-Wasserstein distance, we get

$$\mathcal{W}_2(\mathcal{N}_1, \mathcal{N}_2) = ||\mu_1 - \mu_2||^2 + \mathrm{Tr}\left[(\Sigma_1^{1/2} - \Sigma_2^{1/2})(\Sigma_1^{1/2} - \Sigma_2^{1/2})^T + \Sigma_1^{1/2}\Sigma_2^{1/2}\right.$$
$$\left. + \Sigma_2^{1/2}\Sigma_1^{1/2} - 2(\Sigma_2^{1/2}\Sigma_1\Sigma_2^{1/2})^{1/2}\right] \quad (5)$$

We proceed by noting that the Trace operator is linear; implying $\mathrm{Tr}(A + B) = \mathrm{Tr}(A) + \mathrm{Tr}(B)$, allowing us to analyse the terms separately. Next, we note that the Frobenius norm of a matrix is related to its trace by: $||A||_F^2 = \mathrm{Tr}(AA^T)$. Therefore,

$$\mathrm{Tr}\left[(\Sigma_1^{1/2} - \Sigma_2^{1/2})(\Sigma_1^{1/2} - \Sigma_2^{1/2})^T\right] = ||\Sigma_1^{1/2} - \Sigma_2^{1/2}||_F^2\,. \quad (6)$$

Since the Trace operator is cyclic; implying $\mathrm{Tr}(AB) = \mathrm{Tr}(BA)$, we have

$$\mathrm{Tr}(\Sigma_1^{1/2}\Sigma_2^{1/2} + \Sigma_2^{1/2}\Sigma_1^{1/2}) = 2\mathrm{Tr}(\Sigma_1^{1/2}\Sigma_2^{1/2}) \quad (7)$$

Substituting Eqs. 6 and 7 into Eq. 5, we have

$$\mathcal{W}_2(\mathcal{N}_1, \mathcal{N}_2) = ||\mu_1 - \mu_2||^2 + ||\Sigma_1^{1/2} - \Sigma_2^{1/2}||_F^2 + 2\mathrm{Tr}\left[\Sigma_1^{1/2}\Sigma_2^{1/2} - (\Sigma_2^{1/2}\Sigma_1\Sigma_2^{1/2})^{1/2}\right] \quad (8)$$

We note that in the trivial case where $\Sigma_1$ and $\Sigma_2$ are commutative, the trace is reduced to zero. However, what happens in the general case when the covariance matrices are not commutative? We address this through proposition 1, which shows that

$$\mathrm{Tr}\left[(\Sigma_2^{1/2}\Sigma_1\Sigma_2^{1/2})^{1/2}\right] \geq \mathrm{Tr}(\Sigma_2^{1/2}\Sigma_1^{1/2})\,.$$

Therefore, on substitution the trace terms cancel out, leading to a familiar expression wrapped in an inequality:

$$\mathcal{W}_2(\mathcal{N}_1, \mathcal{N}_2) \leq ||\mu_1 - \mu_2||^2 + ||\Sigma_1^{1/2} - \Sigma_2^{1/2}||_F^2 \quad (9)$$
$$\square$$

**Proposition 1.** *Let $A, B$ be any two positive definite matrices not necessarily commutative. Then,*

$$Tr\left[(A^{1/2}BA^{1/2})^{1/2}\right] \geq Tr(A^{1/2}B^{1/2})$$

*Proof.* Let $X = A^{1/2}B^{1/2}$, consequently we need to prove that $\mathrm{Tr}\left[(XX^T)^{1/2}\right] \geq \mathrm{Tr}(X)$. Let $X = PU$ be the polar decomposition of $X$ where $P$ is a positive definite matrix and $U$ is a orthogonal matrix since $X$ is a real matrix. On substitution, we have $\mathrm{Tr}\left[((PU)(PU)^T)^{1/2}\right] \geq \mathrm{Tr}(PU)$. Since P by definition is symmetric and U is orthogonal, we need to show that $\mathrm{Tr}\left[P\right] \geq \mathrm{Tr}(PU)$.

Let $P = Q\Lambda Q^T$ be the eigendecomposition of P where $Q$ is the orthonormal basis and $\Lambda$ is the diagonal matrix consisting of the eigenvalues $\lambda_i$. Since the trace is the sum of eigenvalues of a matrix, we need to show that $\sum_i \lambda_i \geq \mathrm{Tr}(Q\Lambda Q^T U)$. Since the trace is cyclic, we have $\sum_i \lambda_i \geq \mathrm{Tr}(\Lambda Q^T UQ) = \mathrm{Tr}(\Lambda\mathcal{K})$, where $\mathcal{K} = Q^T UQ$ is an orthogonal matrix by virtue of being a product of orthogonal matrices. Since $\Lambda$ is a diagonal matrix, $\mathrm{Tr}(\Lambda\mathcal{K}) = \sum_i \lambda_i \mathcal{K}_{ii}$. Moreover, as $\mathcal{K}$ is an orthogonal matrix, $\mathcal{K}_{ii} <= 1$. As a consequence, $\sum_i \lambda_i \geq \sum_i \lambda_i \mathcal{K}_{ii}$, concluding our proof. $\square$

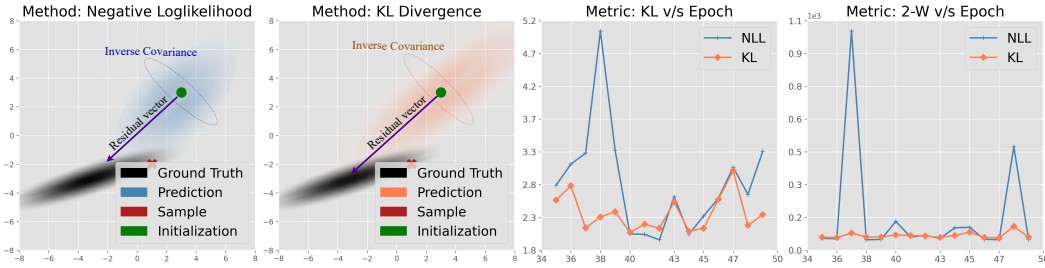

Figure 7: *Residuals and sub-optimal convergence (Section: 3.1).* The residual can be treated as a vector which approximately points along the line segment joining the predicted mean and the target mean. However, if the residual is large, we observe that it influences the predicted covariance significantly. Consequently, the inverse covariance is aligned orthogonal to the residual vector. Since the gradient of the mean estimator is directly proportional to the inverse (Eq. 11), the gradient as a whole is desensitised to move towards the target. In fact, the inverse magnifies gradient updates in the direction orthogonal to the residual vector, potentially leading to oscillations.

## A.2 PROOF OF LEMMA 1

*Proof.* The optimal solution for $\widehat{\Sigma}_Y(X)$ involves minimizing

$$\sum_{i=1}^N D_{\mathrm{KL}}(\mathcal{N}(\boldsymbol{y}_i, \Sigma_Y^{(\mathrm{prior})}(X)) \,||\, \mathcal{N}(\widehat{\mu}_Y(X), \widehat{\Sigma}_Y(X))).$$

Using the definition of the KL Divergence (Eq. 2) and dropping the non-parametric terms, we get

$$\frac{1}{N}\sum_{i=1}^N \frac{1}{2}\left[\mathrm{Tr}(\widehat{\Sigma}_Y^{-1}(X)\Sigma_Y^{(\mathrm{prior})}(X)) + (\widehat{\mu}_Y(X) - \boldsymbol{y}_i)^\top \widehat{\Sigma}_Y^{-1}(X)(\widehat{\mu}_Y(X) - \boldsymbol{y}_i) - \ln|\widehat{\Sigma}_Y^{-1}(X)|\right] \quad (10)$$

Setting the derivative *w.r.t* the predicted mean $\widehat{\mu}_Y(X)$ to 0, we get

$$\frac{1}{N}\sum_{i=1}^N \widehat{\Sigma}_Y^{-1}(X)(\widehat{\mu}_Y(X) - \boldsymbol{y}_i) = 0 \tag{11}$$

$$\widehat{\mu}_Y(X) = \frac{1}{N}\sum_{i=1}^N \boldsymbol{y}_i \tag{12}$$

This is indeed the same solution as minimizing the negative log-likelihood, and therefore $\widehat{\mu}_Y(X)$ predicts the correct mean. However, setting the derivative *w.r.t* the predicted precision $\widehat{\Sigma}_Y^{-1}(X)$ gives us three terms of the form (1) $\mathrm{Tr}(AB)$, (2) $b^T A b$ and (3) $\ln|A|$, where A is the shorthand for precision and B represented different terms. The derivative of the form $\mathrm{Tr}(AB)$ *w.r.t* $A$ is $B^T$ (Eq. 100 in (Petersen & Pedersen, 2012)). Here, $B$ is the prior term $\Sigma_Y^{(\mathrm{prior})}(X)$. Since the prior is symmetric, the derivative of term 1 is $\Sigma_Y^{(\mathrm{prior})}(X)$. The derivative of the form $b^T A b$ is $bb^T$ (Eq. 72 in (Petersen & Pedersen, 2012)). Therefore, the derivative of term 2 is $(\widehat{\mu}_Y(X) - \boldsymbol{y}_i)(\widehat{\mu}_Y(X) - \boldsymbol{y}_i)^T$. Finally, the derivative of the form $\ln|A|$ is $(A^{-1})^T$ (Eq. 57 in (Petersen & Pedersen, 2012)). Therefore, the derivate of term 3 is $\widehat{\Sigma}_Y(X)$. By combining the three terms, the derivative of Eq. 10 is

$$\frac{1}{N}\sum_{i=1}^N \left[\Sigma_Y^{(\mathrm{prior})}(X) + (\widehat{\mu}_Y(X) - \boldsymbol{y}_i)(\widehat{\mu}_Y(X) - \boldsymbol{y}_i)^T - \widehat{\Sigma}_Y(X)\right] = 0$$

$$\widehat{\Sigma}_Y(X) = \Sigma_Y^{(\mathrm{prior})}(X) + \frac{1}{N}\sum_{i=1}^N (\widehat{\mu}_Y(X) - \boldsymbol{y}_i)(\widehat{\mu}_Y(X) - \boldsymbol{y}_i)^T \tag{13}$$

$$\widehat{\Sigma}_Y(X) \approx \Sigma_Y^{(\mathrm{prior})}(X) + \Sigma_Y(X) \tag{14}$$

$\square$

Note: In comparison, the optimal value of the covariance using the negative log-likelihood is $\widehat{\Sigma}_Y(X) = \frac{1}{N}\sum_{i=1}^N (\widehat{\mu}_Y(X) - \boldsymbol{y}_i)(\widehat{\mu}_Y(X) - \boldsymbol{y}_i)^T$

## B  EXPERIMENT DETAILS

**Training.** We use separate networks to estimate the mean and covariance, with no overlapping parameters, following the results of Stirn et al. (2023). This is also advocated for by Sluijterman et al. (2024). The architectures of these network are described in the different subsections. The mean and covariance have the same architecture with the exception of the final layer. We use the pseudo-labels $\widetilde{\Sigma}_Y(\boldsymbol{x})$ together with the input $\boldsymbol{x}$ and target $\boldsymbol{y}$ to train the mean and covariance estimators simultaneously using the 2-Wasserstein bound. Hence, instead of having training pairs $(\boldsymbol{x}, \boldsymbol{y})$, we have triplets $(\boldsymbol{x}, \boldsymbol{y}, \widetilde{\Sigma}_Y(\boldsymbol{x}))$. Unless specified, we do not use warm-up in our experiments. Moreover, the bound does not require warm-up since the mean and covariance estimator training is decoupled. All the methods are trained using the AdamW optimizer, which implicitly imposes a weight decay of 0.01 on the parameters. We ensure a fair comparison by randomly initializing all methods with the same mean and covariance estimators, and each method uses its own learning rate scheduler. Additionally, the batching and sample ordering are the exact same across all methods. At training time, we sample a batch which is simultaneously used by all the baselines for optimization.

### B.1  SYNTHETIC DATA

**Univariate.** We draw 50,000 samples for each of the three different sinusoidal distributions: (1) $y = |x| \sin(2\pi x)$ (2) $y = (5 - |x|) \sin(2\pi x)$ (3) $y = 5 \sin(2\pi x)$, all of them with heteroscedastic noise $\sigma(x) = |x|$ (2). We train a fully connected feed-forward neural network with batch normalization (Ioffe & Szegedy, 2015) and *tanh()* activation to learn the mean and variance. Specifically, we use four hidden layers with a latent dimension of fifty. Every alternate layer is followed by the batch normalization layers. The full results are shown in Fig. 10.

**Multivariate.** The dimensionality of the input and target, $\boldsymbol{x}$ and $\boldsymbol{q}$ is varied from 4 to 32 in steps of 4, and the mean and standard deviation are reported over ten trials for each dimension. Depending on the dimensionality, between 4000 and 20000 samples are drawn. Similar to our univariate setup, we train a fully connected feed-forward neural network with batch normalization but *ELU()* activation to learn the mean and covariance. Specifically, we use ten hidden layers with a latent dimension of that is the dimensionality of the input squared. The idea is that the size of the network increases as the dimensionality of the network increases to account for increasing complexity. Every alternate layer is followed by the batch normalization layers. The full results are shown in Figure 11. We report the computational requirements in Table 1.

### B.2  UCI REGRESSION

We follow the experimental setup in Shukla et al. (2024). For each of the twelve datasets, 25% of the features are randomly selected as inputs, with the remaining 75% used as multivariate targets during run-time. Although some of the resulting input-target pairings may yield sub-optimal performance in prediction, this presents a valuable test for the covariance estimator, which needs to identify correlations even in challenging scenarios. Moreover, the random assignment of features guarantees that our experiments are unbiased, as the selection process is not manipulated. All datasets are standardized to a mean of zero and a variance of one. We reuse our neural network architecture from the multivariate experiments. We perform five trials for each dataset and report the mean and standard deviation in Table 3.

### B.3  2D HUMAN POSE ESTIMATION

ViTPose (Xu et al., 2022) is a recent state-of-the-art model that adapts vision transformers (Dosovitskiy et al., 2021) for the task of human pose estimation. We use the base version (ViTPose-B) for our task Additionally, we use soft-argmax (Li et al., 2021b;a) which is applied to reduce the heatmap, initially a tensor of shape $N \times 64 \times 64$, to a 1D vector of length $2N$, where $N$ is the number of joints in the human pose. To obtain the input for the covariance estimator, we use residual connections which involves downscaling and upscaling of the 1-D features predicted by the backbone network. The output of the downscaling is used to predict the covariance. We perform our experiments on the MPII (Andriluka et al., 2014) and LSP/LSPET (Johnson & Everingham, 2010; 2011) datasets, with the latter focusing on poses related to sports. We merge the MPII and LSP-LSPET

datasets to increase the sample size. The pose estimator is trained using the Adam optimizer with a 'ReduceLROnPlateau' learning rate scheduler for 100 epochs, with the learning rate set to 1e-3. Two augmentations, Shift+Scale+Rotate and horizontal flip, are applied. For details on the specific implementation, readers are referred to the code.

## C COMPILATION OF RESULTS

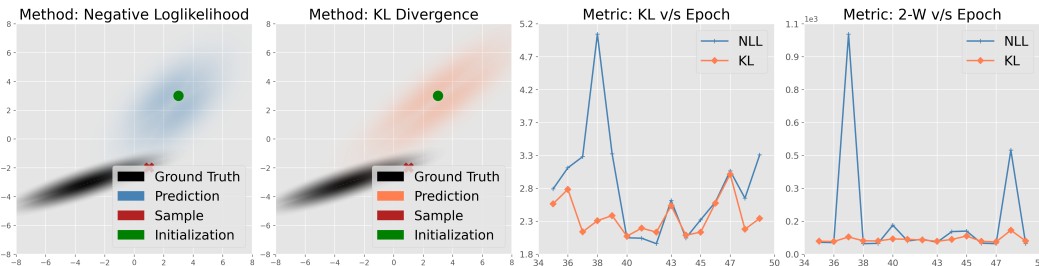

(a) We observe that the KL Divergence can act as a regularizer over the learnt covariance, thereby stabilizing optimization. However, the covariance for both the methods is dominated by the residual term, slowing down convergence.

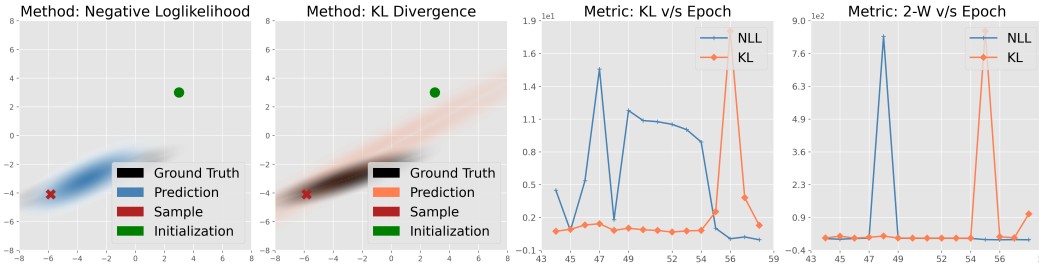

(b) At a higher learning rates (1e-1), both the Negative Log-Likelihood and the KL Divergence oscillate around the true distribution resulting in unstable optimization.

Figure 8: **Bivariate Normal Distribution (A)** *Impact of residuals in optimization (Section: 3.1).* In addition to feature granularity (Seitzer et al., 2022), we show that a source for subpar convergence arises from the susceptibility of the negative log-likelihood and KL-Divergence to residuals in optimization.

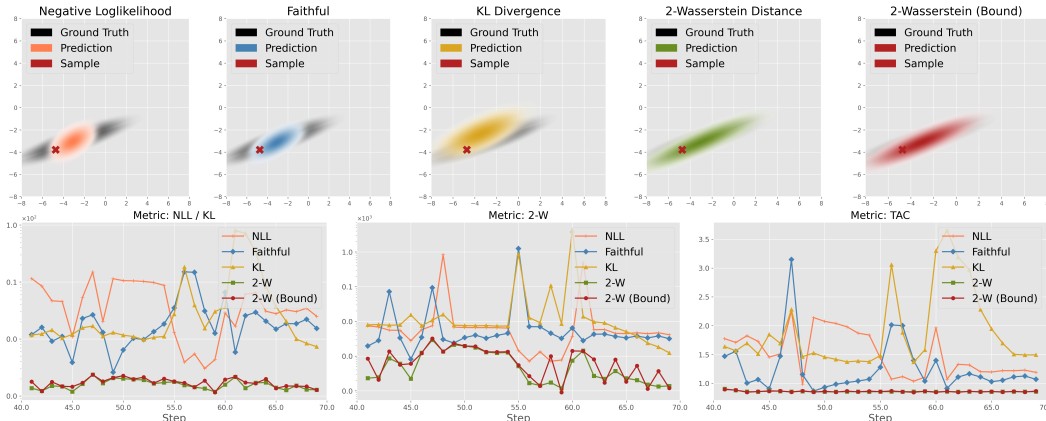

(a) We observe that the KL-Divergence and likelihood based methods: vanilla negative log-likelihood and faithful (Stirn et al., 2023) result in unstable convergence. In comparison, the 2-Wasserstein based methods are much more stable and accurate since they are not affected by residuals nor is the covariance affected by the convergence of the mean estimator.

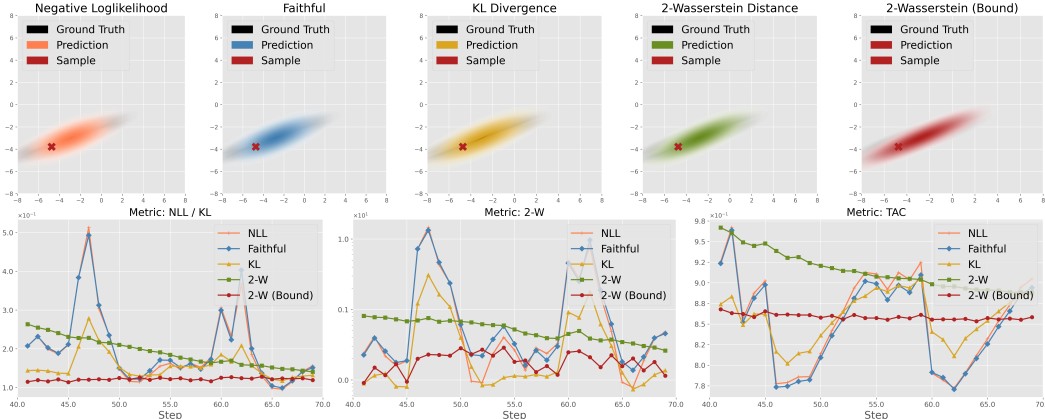

(b) If we initialize the predicted mean to the true mean, we still observe unstable convergence for the divergence and likelihood based methods due to perturbations caused by residuals (Section: 3.1).

Figure 9: **Bivariate Normal Distribution (B)** *Visualizing convergence in bivariate regression (Section: 3.2).* We perform analysis on two settings (a) the mean and covariance of the predicted distribution are initialized away from the target (b) the mean of the predicted distribution is initialized at the mean of the target.

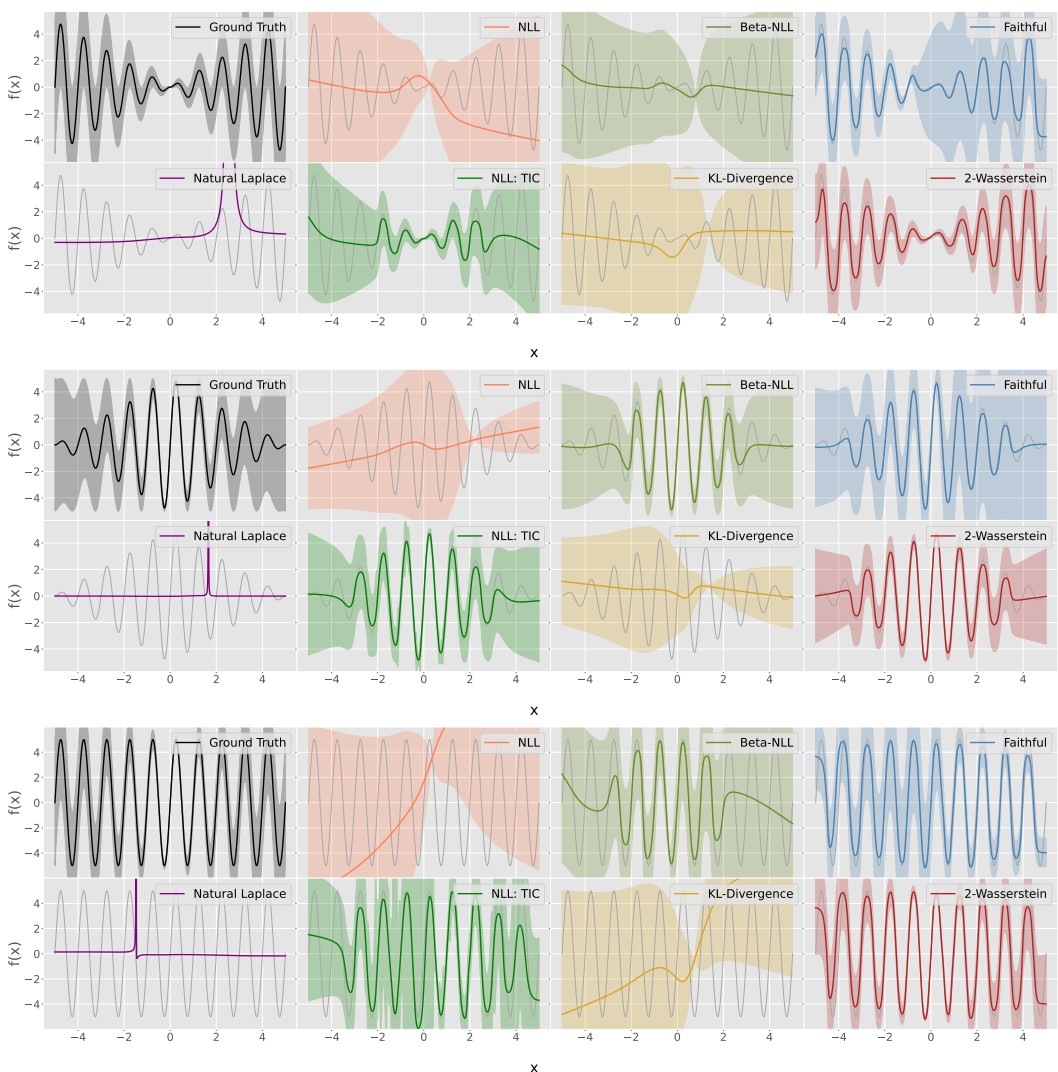

Figure 10: **Univariate.** We define the ground truth sinusoidals as (Top) $y = |x| \sin(2\pi x)$ with $\sigma(x) = |x|$ (Middle) $y = (5 - |x|) \sin(2\pi x)$ and $\sigma(x) = |x|$ (Bottom) $y = 5 \sin(2\pi x)$ and $\sigma(x) = |x|$. Given samples from the ground truth, the networks are trained to learn the underlying distribution using different objectives.

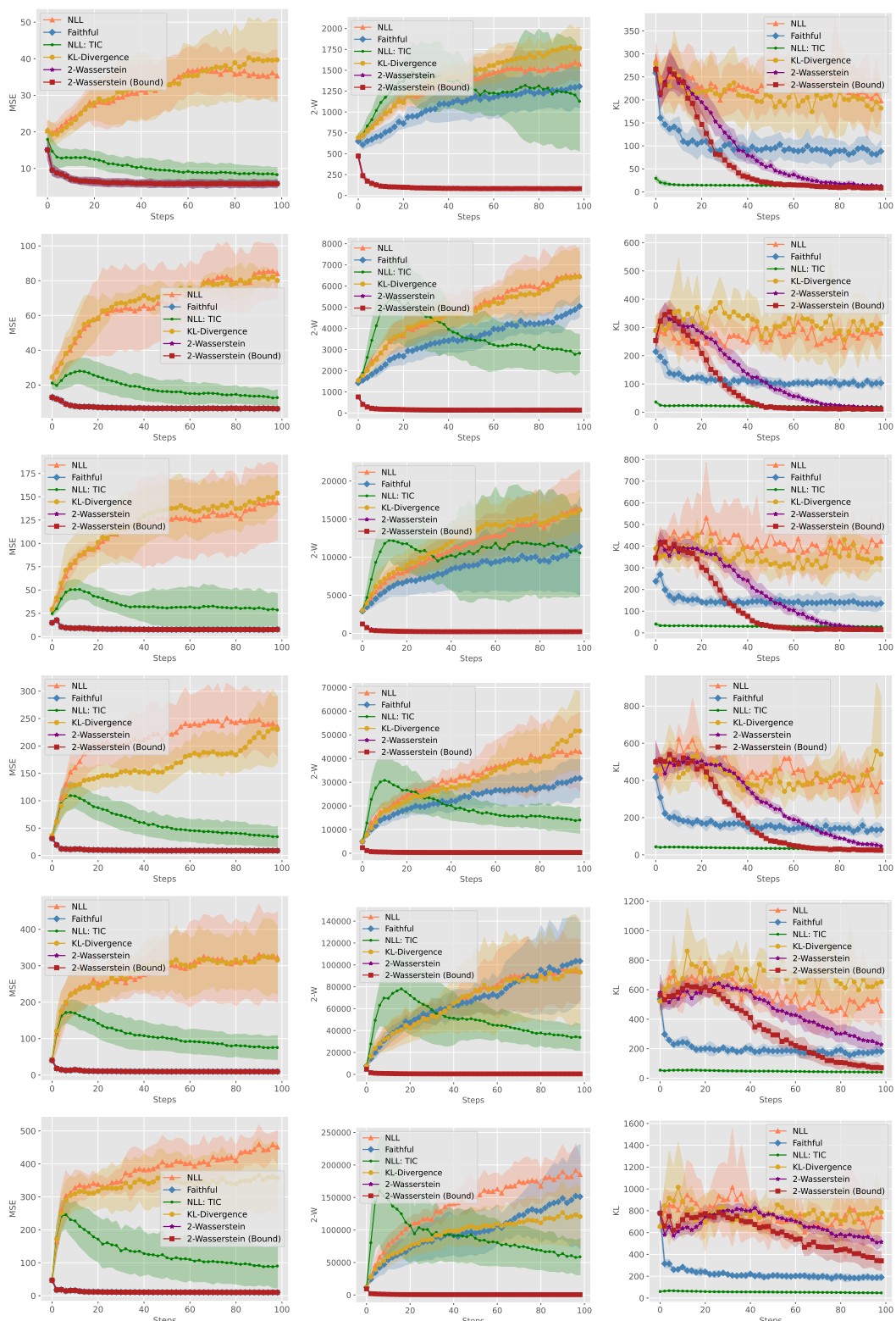

Figure 11: (**Multivariate**). We simulate multivariate data with increasing dimensionality (top row: 12, bottom row: 32, middle rows: increment of four ). An increase in dimensionality causes the mean estimator to converge slow (or diverge) for likelihood and KL-Divergence. The 2-Wasserstein bound is succesfully able to learn the mean and covariance across all dimensions.

Table 3: **UCI Regression.** Results Across Different Metrics: Mean Square Error (MSE), Task Agnostic Correlations (TAC), and Negative Log-Likelihood (NLL) with standard deviations.

**Mean Square Error (MSE)**

| Method | Abalone | Air | Appliances | Concrete | Electrical | Energy | Gas | Naval | Parkinson | Power | Red Wine | White Wine |
|---|---|---|---|---|---|---|---|---|---|---|---|---|
| NLL | 3.74 ± 1.65 | 17.92 ± 5.23 | 53.49 ± 24.10 | 4.57 ± 1.07 | 9.28 ± 3.65 | 4.20 ± 1.75 | 10.98 ± 8.35 | 10.34 ± 5.07 | 54.51 ± 23.94 | 9.09 ± 2.67 | 8.94 ± 4.53 | 9.40 ± 4.95 |
| KL-Divergence | 1.90 ± 1.17 | 14.70 ± 3.63 | 90.90 ± 20.79 | 3.84 ± 1.04 | 15.57 ± 11.22 | 4.20 ± 0.84 | 10.16 ± 5.28 | 12.39 ± 4.62 | 59.39 ± 15.98 | 9.97 ± 2.99 | 7.26 ± 2.39 | 8.17 ± 2.33 |
| Beta-NLL | 0.35 ± 0.13 | 1.58 ± 0.83 | 3.69 ± 1.93 | 2.02 ± 0.57 | 3.62 ± 2.63 | 1.87 ± 0.99 | 1.50 ± 0.33 | 0.72 ± 0.55 | 8.11 ± 8.25 | 3.06 ± 1.10 | 2.15 ± 0.64 | 3.43 ± 1.97 |
| NLL: Diagonal | 1.32 ± 0.22 | 8.90 ± 2.03 | 37.91 ± 6.39 | 4.28 ± 1.82 | 6.58 ± 1.34 | 3.99 ± 1.06 | 5.73 ± 1.26 | 9.60 ± 5.12 | 27.35 ± 5.68 | 6.52 ± 1.97 | 5.75 ± 2.26 | 6.01 ± 1.35 |
| Faithful | **0.16 ± 0.03** | **0.33 ± 0.03** | **0.20 ± 0.03** | **0.72 ± 0.08** | **0.89 ± 0.07** | **0.41 ± 0.18** | **0.45 ± 0.13** | **0.06 ± 0.05** | **0.29 ± 0.08** | **0.61 ± 0.10** | **0.70 ± 0.05** | **0.78 ± 0.05** |
| NLL: TIC | 0.21 ± 0.04 | 0.82 ± 0.37 | 4.45 ± 3.78 | 0.96 ± 0.25 | 0.91 ± 0.07 | 0.61 ± 0.14 | 0.67 ± 0.37 | 1.36 ± 0.41 | 8.89 ± 6.46 | 0.66 ± 0.08 | 0.97 ± 0.34 | 0.92 ± 0.12 |
| **2-W (Bound)** | **0.16 ± 0.03** | **0.34 ± 0.03** | **0.20 ± 0.03** | **0.72 ± 0.08** | **0.90 ± 0.18** | **0.41 ± 0.18** | **0.45 ± 0.13** | **0.07 ± 0.04** | **0.30 ± 0.08** | **0.61 ± 0.10** | **0.71 ± 0.04** | **0.79 ± 0.05** |

**Task Agnostic Correlations (TAC)**

| Method | Abalone | Air | Appliances | Concrete | Electrical | Energy | Gas | Naval | Parkinson | Power | Red Wine | White Wine |
|---|---|---|---|---|---|---|---|---|---|---|---|---|
| NLL | 3.03 ± 1.26 | 5.21 ± 1.31 | 15.71 ± 4.54 | 2.94 ± 0.65 | 5.74 ± 1.41 | 3.15 ± 0.78 | 3.85 ± 1.69 | 3.36 ± 1.17 | 14.07 ± 3.16 | 4.72 ± 1.22 | 4.32 ± 1.10 | 4.88 ± 1.06 |
| KL-Divergence | 1.90 ± 0.73 | 7.07 ± 1.91 | 20.04 ± 4.72 | 3.60 ± 1.07 | 6.32 ± 1.82 | 3.12 ± 0.50 | 3.76 ± 0.78 | 5.16 ± 2.43 | 13.97 ± 2.41 | 4.49 ± 0.88 | 4.93 ± 0.85 | 4.28 ± 1.09 |
| Beta-NLL | 0.40 ± 0.08 | 0.81 ± 0.18 | 1.05 ± 0.15 | 1.03 ± 0.14 | 1.31 ± 0.32 | 0.98 ± 0.29 | 0.79 ± 0.06 | 0.41 ± 0.12 | 1.34 ± 0.46 | 1.18 ± 0.16 | 1.03 ± 0.15 | 1.17 ± 0.29 |
| NLL: Diagonal | 0.87 ± 0.08 | 2.15 ± 0.15 | 4.09 ± 0.38 | 1.50 ± 0.29 | 1.94 ± 0.17 | 1.45 ± 0.16 | 1.78 ± 0.20 | 2.08 ± 0.61 | 3.53 ± 0.36 | 1.90 ± 0.29 | 1.66 ± 0.33 | 1.83 ± 0.20 |
| Faithful | 0.54 ± 0.07 | 1.31 ± 0.17 | 1.51 ± 0.17 | 1.48 ± 0.33 | 2.39 ± 0.07 | 1.21 ± 0.07 | 1.16 ± 0.13 | 1.54 ± 0.17 | 1.54 ± 0.17 | 1.61 ± 0.20 | 1.65 ± 0.14 | 1.81 ± 0.22 |
| NLL: TIC | 0.29 ± 0.02 | 0.53 ± 0.04 | 0.44 ± 0.10 | 0.70 ± 0.16 | **0.70 ± 0.04** | 0.59 ± 0.10 | **0.43 ± 0.10** | 0.25 ± 0.04 | 0.61 ± 0.10 | **0.51 ± 0.03** | 0.79 ± 0.10 | 0.67 ± 0.11 |
| **2-W (Bound)** | **0.24 ± 0.02** | **0.34 ± 0.02** | **0.24 ± 0.01** | **0.51 ± 0.02** | 0.67 ± 0.01 | **0.36 ± 0.02** | 0.35 ± 0.02 | **0.12 ± 0.04** | **0.28 ± 0.03** | 0.50 ± 0.03 | **0.49 ± 0.01** | **0.56 ± 0.01** |

**Negative Log-Likelihood (NLL)**

| Method | Abalone | Air | Appliances | Concrete | Electrical | Energy | Gas | Naval | Parkinson | Power | Red Wine | White Wine |
|---|---|---|---|---|---|---|---|---|---|---|---|---|
| NLL | 35.89 ± 28.55 | 56.98 ± 9.73 | 245.99 ± 73.55 | 28.50 ± 6.62 | 63.15 ± 15.57 | 29.85 ± 8.99 | 41.03 ± 21.62 | 38.18 ± 5.01 | 262.59 ± 73.52 | 49.37 ± 21.28 | 46.58 ± 10.26 | 58.06 ± 16.90 |
| KL-Divergence | 18.27 ± 4.74 | 83.18 ± 25.55 | 413.96 ± 151.56 | 38.23 ± 17.81 | 73.87 ± 23.77 | 28.50 ± 4.81 | 34.38 ± 6.96 | 49.24 ± 23.79 | 257.36 ± 92.79 | 45.49 ± 11.37 | 61.96 ± 24.34 | 48.66 ± 16.96 |
| Beta-NLL | 9.80 ± 1.17 | 29.38 ± 2.89 | 60.30 ± 1.67 | 20.45 ± 4.74 | 35.44 ± 4.59 | 20.15 ± 5.27 | 20.26 ± 1.43 | 20.81 ± 2.21 | 59.98 ± 11.96 | 27.64 ± 4.56 | 34.05 ± 5.60 | 30.95 ± 2.79 |
| NLL: Diagonal | 18.61 ± 5.96 | 80.86 ± 8.46 | 369.67 ± 88.48 | 46.82 ± 15.19 | 65.73 ± 10.47 | 36.09 ± 8.57 | 47.06 ± 15.98 | 77.47 ± 45.51 | 238.71 ± 38.34 | 51.60 ± 3.45 | 77.98 ± 17.61 | 67.21 ± 16.24 |
| Faithful | 11.86 ± 1.18 | 33.31 ± 1.53 | 65.15 ± 2.49 | 17.42 ± 1.68 | 34.73 ± 0.53 | 19.41 ± 0.49 | 22.47 ± 0.86 | 27.70 ± 0.81 | 57.04 ± 3.33 | 24.08 ± 0.87 | 24.34 ± 0.99 | 26.01 ± 1.53 |
| NLL: TIC | **4.71 ± 1.19** | 16.46 ± 0.89 | 30.41 ± 13.42 | 11.36 ± 1.50 | **14.97 ± 0.80** | 12.06 ± 1.44 | **9.96 ± 2.73** | 14.99 ± 3.43 | 42.52 ± 7.52 | **9.31 ± 0.92** | 14.66 ± 1.24 | **12.33 ± 1.68** |
| **2-W (Bound)** | 6.32 ± 0.23 | **13.58 ± 0.25** | **22.72 ± 0.15** | **8.96 ± 0.29** | 15.57 ± 0.24 | **8.85 ± 0.34** | 10.49 ± 0.36 | **11.44 ± 0.18** | **21.48 ± 0.63** | 11.31 ± 0.39 | **11.65 ± 0.17** | 12.12 ± 0.18 |

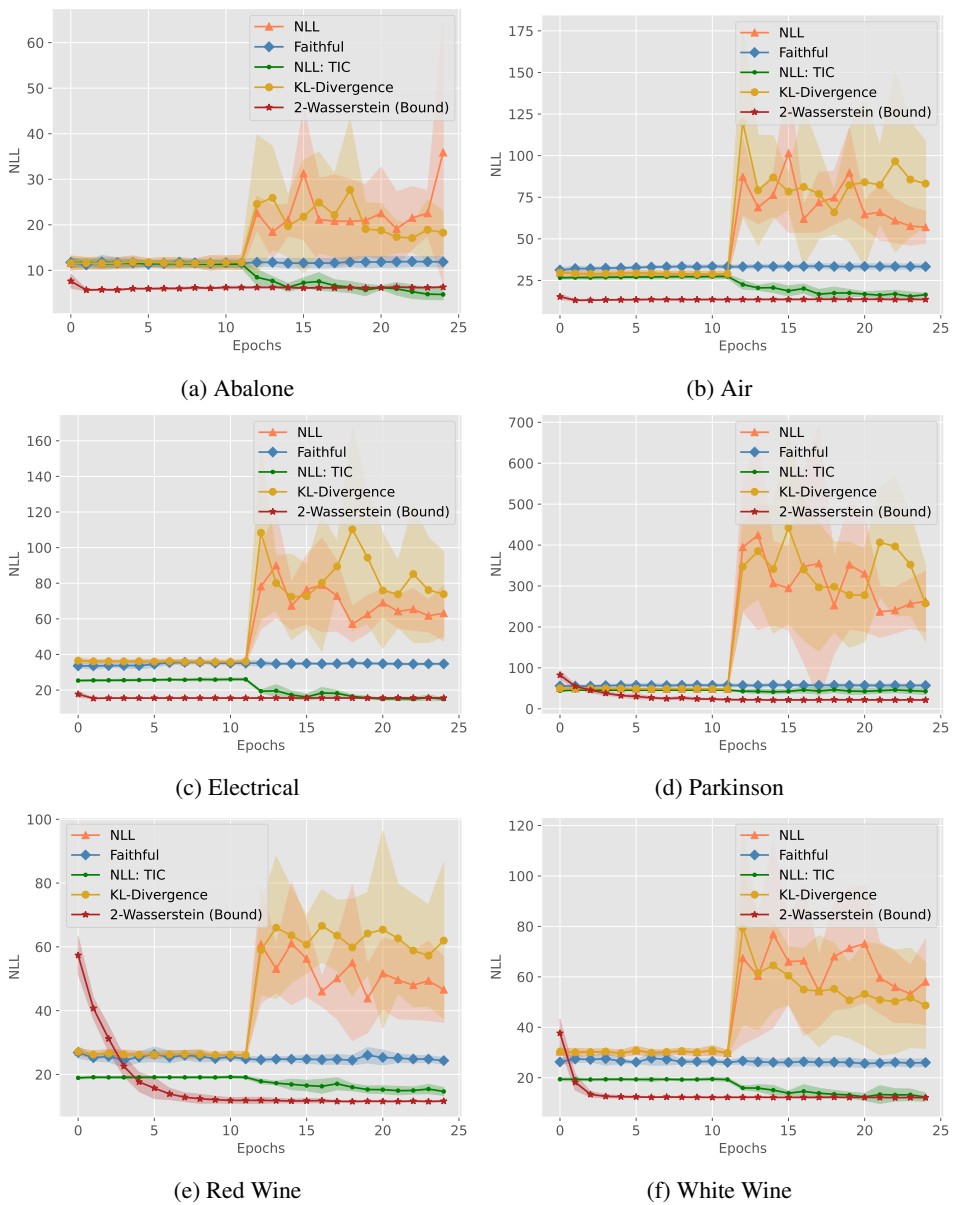

Figure 12: **(Warm-up, UCI)** We explore training deep heteroscedastic regression models using warm-up as proposed in Sluijterman et al. (2024). We train only the mean estimator for half the training epochs, and jointly train the mean and covariance estimator for the remaining half. We observe a trend across datasets that the training is unstable and momentarily diverges for the negative log-likelihood and KL-Divergence which are especially sensitive to residuals and incorrect covariance estimates. This trend is similar to our observations for the two methods on our bivariate normal distribution experiments (Fig. 9). While we plot for six datasets here, this trend is representative of all datasets.

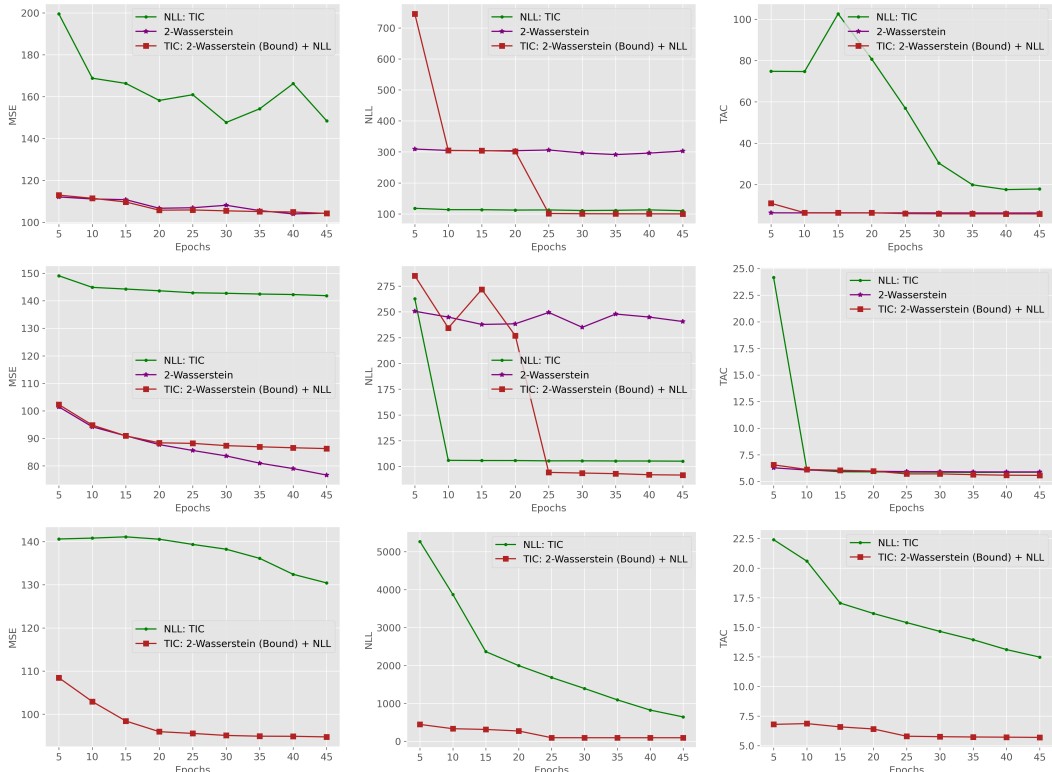

Figure 13: **Human Pose**. *(Improving state-of-the-art heteroscedastic pose estimation)* (Top row: learning rate: 1e-2, Middle row: learning rate: 1e-3, Bottom row: learning rate: 1e-4) We explore a hybrid training strategy by combining the 2-Wasserstein bound with the negative log-likelihood. We train ViTPose for the first 20 epochs using the bound, and then switch to negative log-likelihood. We use the TIC parameterization for the covariance which when trained with the negative log-likelihood, showed state-of-the-art performance in heteroscedastic pose estimation. We observe that using the hybrid approach retains the competitiveness of the 2-Wasserstein bound on the mean square error, and the competitiveness of the negative log-likelihood on the negative log-likelihood metric.

