# OpenReview forum: "Towards Self-Supervised Covariance Estimation in Deep Heteroscedastic Regression"
_ICLR.cc/2025/Conference — ICLR 2025 Poster_

### Official Review · Reviewer_ZXre · 2024-11-02

**Soundness:** 3
**Presentation:** 3
**Contribution:** 2
**Rating:** 6
**Confidence:** 3

**Summary:**

The paper under review is concerned with the problem of Deep Heteroscedastic Regression in the case where the variance is known. In the classical problem, given samples from the joint distribution of $(X,Y)$, one aims to estimate the mean and covariance matrix of the conditional distribution $Y|X$ as a function of $x$ using maximum likelihood estimation. The authors consider the case where a reference distribution is known and consider two distance measures, the KL divergence and the 2-Wasserstein metric between the distribution to be optimized (given by mean and covariance) and a distribution with known covariance. The authors then also consider the problem of how to obtain pseudo-labels in the absence of a ground truth. The paper concludes with extensive experiments, suggesting that using the 2-Wasserstein distance coubpled with preudo-label generation results in a competitive performance.

**Strengths:**

The paper is generally well written, and apart from a few minor mistakes indicated below, the mathematical derivations are correct and sound. The graphics are informative and well readable. The results on synthetic and real datasets look convincing, though in the absence of the code (which is due to be published) I can't verify everything.

**Weaknesses:**

The problem formulation is not clear. The authors could start by 1) specifying the terminology they use. For example, what is the 'target'? What is the data given and what do we want to get from the data? What are the precise assumptions on the distribution? The preliminaries section in Seitzer et al. 2022 is a good example on how to set the stage (incidentally, you could call the mean $\mu$, which seems more common). There are a few typos and some minor mistakes in the mathematical derivations (see questions below). The appendix on experiments seems to have been written a bit in a hurry (there are some typos, like capitalizing 'univariate').

**Questions:**

* Could you comment on how your approach relates to the literature on normalizing flows?
* Could you specify how the KL-divernce minimization problem is formulated? Since the KL divergence is not symmetric, it makes a difference which distribution is used as first argument and which as second. You use the term "forward KL divergence" but I don't see it defined.
* Use consistent notation for the KL divernce (on page 3 you write KL, while on page 4  you write $D_{\mathrm{KL}}$)
* In the proof of Claim 1, can you check whether (2) is correct or whether the $\ln(\cdot)$ term should be outside of the square brackets? My calculation in the univariate special case indicates that the logarithmic term should not have a factor $1/2$. This also makes the factor $2$ disappear in the conclusion.

---

> ### Author Response · Authors · 2024-11-20
> **Author Response To Reviewer ZXre: Part 1/1**
>
> Dear Reviewer ZXre,
>
> We thank you for your review and your suggestions which we have incorporated in the draft. We hope to address your concerns below:
>
> **W1 - Problem formulation not clear.**: We accept your suggestion and devote a paragraph explaining the problem formulation and notation in the paragraph Preliminaries, Section 2. We also switch our notation to maintain consistency where possible with Seitzer et al. Please let us know if any ambiguities persist.
>
> **W2 - Typos and minor mistakes in the derivation**: We thank you for going through our derivation! We have cross checked Eq. (2) using [b, c]. While possible, we think it is unlikely that the equation or the derivation is incorrect since it tallies with experimental evaluation.
>
> **W3 - Appendix**: We have organised the appendix, and would like to thank you for spotting the errors!
>
> **Q1 - Normalizing Flows**: We do not see a direct connection between the ideas proposed in this work and normalizing flows. If we recall correctly, normalizing flows are a class of generative models which construct a series of invertible transformations to map from a known distribution to an arbitrary distribution. While it may be possible to create applications of Lemma 1 and Theorem 1 in normalizing flows, this is not the scope of our work.
>
> **Q2 - Forward KL Divergence**: We define the forward KL Divergence $D_{KL}(p || q)$ in Section 3.1, specifically Eq. 2. Here, $p$ corresponds to the target (or true) distribution and $q$ corresponds to the predicted distribution. This form of the KL Divergence also gives rise to other optimization measures such as the negative log-likelihood ($-\mathbb{E}_p \log q$).
>
> **Q3 - Consistent Notation**: Thank you for this feedback. We have incorporated this in our draft!
>
> [b] Zhang et al. "*On
> the properties of kullback-leibler divergence between multivariate gaussian distributions.*", NeurIPS 2024
>
> [c] Joram Soch. Kullback-leibler divergence for the multivariate normal distribution, May 2020. https://statproofbook.github.io/P/mvn-kl.html.

---

> > ### Comment · Reviewer_ZXre · 2024-11-22
> >
> > Thank you, the presentation is now clearer and the score for the presentation has been upgraded.

---

> > > ### Author Response · Authors · 2024-11-22
> > >
> > > Thank you for the reply! We wanted to confirm that we have addressed all of your concerns. If yes, we hope you would reconsider your overall score; nevertheless thank you for your positive evaluation!

---

### Official Review · Reviewer_5LBD · 2024-11-03

**Soundness:** 3
**Presentation:** 3
**Contribution:** 3
**Rating:** 8
**Confidence:** 4

**Summary:**

This work studies the problem of estimating the covariance of a heteroscedastic random variable, i.e., a random variable $y$ whose covariance is a function of an associated covariate $x$. (in contrast to the usual homoscedastic case, where the variance of $y$ is independent of $x$). Existing approaches tackle this problem as an unsupervised problem that minimizes a KL-like loss, which tend to be unstable or slow. In this work, the authors study how to better supervise this problem, and how to obtain sensible pseudo labels to solve a self-supervised problem instead. After proposing a simple-to-optimize bound based on the 2-Wasserstein distance, and a way of collecting pseudo-labels by assuming x-continuity of the variance, the authors show that their approach is able to model heteroscedastic cases well and significantly quicker than previous methods.

**Strengths:**

- **S1.** Properly modelling heteroscedastic random variables is an important and often overlooked problem.
- **S2.** The paper is generally well written, and contains enough explanations and intuitions to help the reader.
- **S3.** The experiments are convincing and show the points the authors were trying to make.
- **S4.** The arguments and derivations that precede the proposed methods are clear and convincing.

**Weaknesses:**

- **W1.** The particularities specific to the heteroscedastic setting should be better stressed. For example, I understand that replacing $\mu_p$ by $y_i$ in the claim 1 is done since the variance $\Sigma_y$ depends on the sample $i$ only, but this is not clear.
  - I would also double-check and explicitly write the derivations of Eq. 3, as I am not sure that they are correct.
- **W2.** The derivations are limited to the case of Gaussian predictions. A common assumption, but not clear from the abstract.
- **W3.** The experiments lack standard deviations, yet they were repeated 5 times each.
- **W4.** It is strange that for the last experiment the authors proposed a hybrid approach that combines 3 different methods.
   - It would be at least nice to see what happens if the TIC parametrization is dropped.
   - And, since speed and low-memory were a selling point of the proposed method, it would be necessary to see the penalty in time/memory that the TIC parametrization incurs.
   - It would only be fair to compare with TIC-TAC in this experiment too, as the authors are using half of their contribution.
   - Finally, I would love to see the proposed method applied here with the pseudo-labels taken from the input images themselves.
- **W5.** Some figures are hard to read. I would encourage the authors to consider using a log-scale for the y-axis.

**Questions:**

- **Q1.** I am struggling to understand why the W2-bound is better. Are you parametrizing $\Sigma_1^{1/2}$ directly, so you do not need to compute that square root?
- **Q2.** What is that memory consumption exactly referring to? Those values look quite large.
- **Q3.** Where are the instabilities mentioned in L482? I couldn't find them in the results.
- **Q4.** What is the TIC parametrization exactly?
- **Q5.** The claim in the lines 210-212 is not clear to me from the results. Could you clarify it?

---

> ### Author Response · Authors · 2024-11-20
> **Author Response To Reviewer 5LBD: Part 1/1**
>
> Dear Reviewer 5LBD,
>
> We thank you for your evaluation of the paper! We have noted your suggestion and have made changes in the draft accordingly. We hope to address your concerns below:
>
> **W1**: We now describe the overall problem setting in Preliminaries, Section 2 of the paper. We separately discuss the formulation of Claim/Lemma 1 in the paragraph preceding it. We have cross-checked and provide an explicit derivation of Claim/Lemma 1 in the appendix. Please let us know if there still exists any ambiguity in the derivation!
>
> **W2**: We have updated the abstract to specify that our analysis is restricted to the multivariate normal distribution.
>
> **W3**: Appendix/Table 3 has been updated with the mean and standard deviations for our results on the UCI Regression datasets. The shaded regions in Figure 11 correspond to the standard devation.
>
> **W4 and Q4**: We merge W4 and Q4 since we believe that they are related. We also believe that there may be some confusion over the scope of this experiment, which we clarify.
>
> The TIC parameterization ([a]) predicts the covariance matrix as a function of the gradient and curvature of the mean estimator $\Sigma(x) = k_1(x) \nabla\widehat{\mu}(x) + k_2(x) \nabla^2\widehat{\mu}(x) + k_3(x)$. Here, $k_1(x), k_2(x), k_3(x)$ are learnable parameters. The authors of [a] show that the TIC parameterization for the covariance when trained using the negative log-likelihood (which is referred to as NLL:TIC) significantly improved upon the standard parameterization.
>
> The experiment on heteroscedastic human pose estimation was introduced in [a] with NLL:TIC as the best performing method. Our preliminary experiments using the 2-Wasserstein failed to match the results of NLL:TIC, which we believe is because our pseudo-labels, which are computed based on a low-dimensional representation of the input images, may not necessarily be accurate. However, we identified an alternate paradigm: training the framework initially using the 2-Wasserstein bound and then switching to NLL:TIC helped retain the advantages across both, the 2-Wasserstein bound and NLL:TIC. Therefore, we highlight this specific experiment not necessarily as one where we are computationally efficient, but one which opens up possibilities of using hybrid approaches to further improve the state-of-the-art (NLL:TIC in this case).
>
> While it is tempting to get pseudo-labels directly using the input images, this is computionally prohibitive: a single colour image with a height and width of 256 when flattened would correspond to a vector of size 196,608. Moreover, the space of images is sparsely populated, making it difficult to extract meaningful patterns. Finally, we would expect a well trained model to be invariant to changes in the image such as background variations. Directly using the images forgoes the invariances learnt by the model.
>
>
>
> [a] Shukla et al. *TIC-TAC: A Framework For Improved Covariance Estimation in Deep Heteroscedastic Regression*, ICML 2024
>
> **W5**: We have updated Figures 2 and 9 by setting the vertical axis to be log-scale.
>
> **Q1**: Yes! The original 2-Wasserstein distance (Eq. 3) has one matrix square root operation that requires gradient and cannot be circumvented. This requires us to use eigendecomposition of the matrix, which may be unstable through backpropagation. The 2-Wasserstein bound does not have a matrix square root operation that requires gradient if we predict the square root of the matrix directly. In addition, the 2-Wasserstein bound is simpler to analyze and more intuitive.
>
> **Q2**: The memory consumption refers to the additional memory needed by each of the methods to compute the forward and backward pass during training.
>
> **Q3**: The instabilties unfortunately occur at random and are hard to tabulate. Therefore, we provide a reference in the paper to the documentation which highlights that this is a known issue due to numerical instability. However, we notice that some datasets such as superconductivity almost always face this instability. Since NLL:TIC also takes a large amounts of time and memory on this dataset, we do not report it in the results.
>
> **Q5**: We thank you for pointing out this ambiguity. We refer you to appendix/Fig. 7: The residual can be treated as a vector which approximately points along the line segment joining the predicted mean and the target mean. However, if the residual is large, we observe that it influences the predicted covariance significantly. Consequently, the inverse covariance is aligned orthogonal to the residual vector. Since the gradient of the mean estimator is directly proportional to the inverse, the gradient as a whole is desensitised to move towards the target. Please let us know if additional clarification is required.

---

> > ### Comment · Reviewer_5LBD · 2024-11-22
> >
> > Dear authors, thank you for the detailed response and for carrying out changes to improve the state of the manuscript. As of now, I am happy to say I have no further questions, but I'll let you know if this changes.

---

> > > ### Author Response · Authors · 2024-11-25
> > >
> > > Thank you once again for your review!

---

### Official Review · Reviewer_6hor · 2024-11-03

**Soundness:** 2
**Presentation:** 3
**Contribution:** 3
**Rating:** 5
**Confidence:** 4

**Summary:**

The paper deals with heteroscedastic regression, and proposes to derive a signal from the neighborhood of a given observation to supervise the covariance. The authors derive pseudo labels for the covariance by looking at the $k$-NNs to a given observation wrt the Malanobis distance. The distance is interpreted in a probabilistic sense and used to compute the expected mean and covariance over the neighboring targets. To integrate the pseudo labels for the covariance in the loss, the authors propose an upper-bound over the 2-Wasserstein distance to be optimized, where the observed target value for each observation and the pseudo label for the covariance allow supervision of both mean and covariance.

**Strengths:**

- The paper deals with the relevant problem of reaching accurate and efficient estimation of both the covariance and mean parameters in deep heteroscedastic regression
- The idea of relying on the neighbourhood of a given observation to obtain a signal to use as supervision for the covariance is sensible and, while this idea was already introduced in [1], the authors propose a more computationally efficient solution.
- The empirical results show that the method improves over alternative baselines.
- The paper is well-structured and easy to follow.

[1] Shukla et al. TIC-TAC: A Framework For Improved Covariance Estimation In Deep Heteroscedastic Regression, ICML 2024.

**Weaknesses:**

- Claim 1, how it is formulated, is confusing to me, and it does not seem like a well-posed setup. In particular, if the true covariance is assumed to be known, it does not appear to make sense to estimate it. Also, if there's a proof, I'd suggest to use "Lemma" instead of "Claim".
- The comparison on UCI regression is in my opinion not exhaustive. Why is the method from [4] not reported on the UCI benchmark despite being SOTA? Also, from the values that the authors obtain for e.g. NLL for UCI regression, which are very different from previous works (e.g. [2, 3, 4]) but much more similar to [1], it looks like they did a similar adaptation to [1], where the authors "adapt the datasets for covariance estimation". This appears to be the case, also given the authors mention that they rescale the data to have variance of ten. Can the authors report the results on the original UCI regression dataset, as used in a large number of previous works?  This allows for transparent comparisons with a large number of previous methods.
- The authors advertise the method as computationally efficient, e.g. compared to [1]. However, do the authors grid-search hyperparameters, like weight decay? I think it would help to make this explicit and compare on overall compute time with other state-of-the-art methods like [4] , where grid-search is not needed due to automatic regularization due to Empirical Bayes.
- Do the authors try different values of $\beta$ for the $\beta$-NLL objective? Or which value is chosen?

[1] Shukla et al. TIC-TAC: A Framework For Improved Covariance Estimation In Deep Heteroscedastic Regression, ICML 2024. [2] Gal et al. Dropout as a Bayesian Approximation: Representing Model Uncertainty in Deep Learning, ICML 2015. [3] Seitzer et al. On the Pitfalls of Heteroscedastic Uncertainty Estimation with Probabilistic Neural Networks, ICLR 2022. [4] Immer et al. Effective Bayesian Heteroscedastic Regression with Deep Neural Networks, NeurIPS 2023.

**Questions:**

- Format of citations is often wrong (missing parenthesis). see lines 152,153.
- Why in e.g. the UCI benchmark the comparison in terms of NLL is relegated to the Appendix, while the MSE results are in the main text? Especially in heteroscedastic regression NLL can be much indicative of overall performance.

---

> ### Author Response · Authors · 2024-11-20
> **Author Response To Reviewer 6hor: Part 1/1**
>
> Dear Reviewer 6hor,
>
> We thank you for your review! We have incorporated your feeback in the revised draft, and we hope to address your concerns below:
>
> **W1**: Let us assume we have an input $x$ with the corresponding label $y$. While we know the label, the neural network needs to be trained to predict the label. We draw a similar analogy with Lemma 1; if we knew the ground truth covariance, how do we train the network to predict it? Lemma 1 addresses a specific setting for training the covariance estimator using the KL Divergence. The goal of this lemma is to present necessary tools needed to supervise the covariance estimator using our pseudo labels for the covariance. We acknowledge that the wording may have been ambiguous, and have updated it. In particular, we believe that Lemma 1, along with Theorem 1, is sufficiently general to be of interest across communities.
>
> **W2**: We have re-run our experiments on the UCI Regression dataset with zero-mean-unit-variance standardization, and report the results in Table 2 and appendix/Table 3, the latter containing all three metrics with mean and standard deviation. We note that our observations and learnings still remain the same.
>
> While [4] is indeed state-of-the-art, the method assumes that the output is univariate, as stated in the introduction section of [4]. This is also seen in the evaluation code used by multiple recent works, including [4]. For instance, [Lines 56 to 64 specify the target to be univariate (link)](https://github.com/aleximmer/heteroscedastic-nn/blob/main/hetreg/uci_datasets.py#L15). Another recent work using UCI Regression makes the same assumption of the targets being univariate [Lines 73 and 74 explictly define univariate targets in their code (link)](https://github.com/ewongtoi/deep-heteroskedastic-regression/blob/master/heteroskedastic_nns/datasets/load_uci/uci_loader.py).
>
> Our work addresses limitations in estimating the covariance, and proposes an accurate yet compute efficient way of modelling the heteroscedasticity through the full covariance matrix. As a result, we follow the experimental protocol of previous works [1] that specifically focus on multivariate predictions. Therefore we believe we should not be penalized for the same. Furthermore, we strongly believe that the experiments are well aligned to evaluate the claim of computationally efficient yet accurate covariance estimation in deep heteroscedastic regression.
>
>
> **W3**: We do not grid-search over the space of hyperparameters, for instance we implement the default weight decay of 0.01 through the AdamW optimizer. Moreover, we use the exact same configuration to ensure fair comparison when training all methods. We compare the compute time with other state-of-the-art methods in Table 1 and 2. The analysis in [4] for regularization is done under the assumption that the target is univariate, thereby making direct comparisons infeasible.
>
> **W4**: We choose $\beta$=0.5 which is the value recommended by the authors of $\beta$-NLL in the conclusion section of the work.
>
> **Q1**: We have corrected our citation format, thank you for bringing it to our notice!
>
> **Q2**: We have moved our NLL results to the main paper. The goal was not to relegate NLL to the appendix. Since we address a direct limitation of [1], we wanted to show improved performance on their proposed metric, TAC, and highlight that the method of [1] has room for improvement in the mean square error evaluation.

---

### Official Review · Reviewer_VkG5 · 2024-11-04

**Soundness:** 3
**Presentation:** 2
**Contribution:** 3
**Rating:** 6
**Confidence:** 4

**Summary:**

Deep heteroscedastic regression models provide both a mean and covariance estimate for each input (typically assuming a Gaussian likelihood). These models are difficult to fit due to overfitting, which can lead to (co)variance estimates collapsing to zero. While recent work has primarily focused on how to perform stable training of these models through adaptations to the objective or tinkering with the training process this paper suggests using pseudo labels to learn the covariance. The authors analyze fitting the covariance with the KL divergence and the 2-Wasserstein distance while using pseudo labels that come from a local heuristic algorithm. To optimize against the 2-Wasserstein distance they derive an upper bound. On synthetic and real-world datasets this method was found to perform well in comparison to recent baselines.

**Strengths:**

- Different approach from much of the recent literature in the space
- Good balance of theoretical and empirical support/motivation for this method
- Will provide resources to reproduce results

**Weaknesses:**

- Related works is nearly identical to [Shukla 2024](https://openreview.net/pdf?id=zdNTiTs5gU)
- Missing some recent literature: [Optimal training of Mean Variance Estimation neural networks](https://www.sciencedirect.com/science/article/pii/S0925231224007008), [Understanding Pathologies of Deep Heteroskedastic Regression](https://openreview.net/pdf?id=n5faLvrsA0)
- Somewhat unclear how this methodology is implemented
- See questions

**Questions:**

- This method of using local information for the variance pseudo labels seems similar in spirit to the local mini-batching in Skafte 2019. Can you comment on how these differ?
- Does this relate to kernel methods?
- What sort of architectures can this be used for? Would the mean and covariance networks share some (all up to the final layer) parameters?
- Could pseudo labels also be incorporated into other existing methods for fitting heteroscedastic regression models?
- Took a few reads a little bit confused on how training works. Is there a need for any warmup period for the mean estimate (as with other methods)? Are mean and covariance learned together from the start? Is any information for the mean relevant in the presence of pseudo labels?
- When training against the negative log likelihood, though fits may be imperfect at least the mean and covariance will be coherent together. Are the situations where the covariance conditional on the mean is nonsensical?
- Do the usage of pseudo labels still make sense/work well when the mean model does not fit well to the data (overfit/underfit)?

---

> ### Author Response · Authors · 2024-11-20
> **Author Response To Reviewer VkG5: Part 1/2**
>
> Dear Reviewer VkG5,
> We thank you for your comments and questions! We hope to address your concerns below:
>
> **W1**: Since we addressed a direct limitation of Shukla et al., we were influenced by their related works. We have updated the related works (Section 2) to include discussion of the work of Sluijterman et al. and Wong-Toi et al.
>
> **W2**: We thank you for bringing these works to our notice, and have referenced them in the paper.
> Sluijterman et al. studies recommendations from Nix et al. which advocates for (1) separate networks for the mean and (co)variance, and (2) warm-up to train heteroscedastic models. We also separate the mean and (co)variance networks following a similar observation made by Stirn et al. We study warm-up using our toy example of bivariate normal distributions in Appendix/Fig. 9 (b). Specifically, we initialize the mean of our predicted distribution at the same location as the target distribution, recreating the convergence of the mean estimator. However, we note that convergence is still unstable since residuals can randomly destabilize the mean and covariance estimators. We discuss this in detail in Section 3. We also explore warm-up using the UCI Regression datasets (Section 4.2 and Figure 12) Our findings indicate that deep heteroscedastic regression is susceptible to residuals, which cannot be remedied by using warm-up.
>
> We enjoyed reading Wong-Toi et al. which studies the training dynamics of deep heteroscedastic regression through phase transitions and have added it in our related works. We believe that the works address parallel problem statements. While Wong-Toi et al. restrict their theoretical analysis to the negative log-likelihood, we study the KL Divergence and the 2-Wasserstein distance. In the process, we prove Lemma 1 and Theorem 1 which we believe are sufficiently general enough to be of interest across communities. Moreover, the tools we introduce help us find accurate yet computationally cheap estimates of the covariance, addressing a direct limitation of prior work. Finally, the nature of our results apply to multivariate outputs, whereas Wong-Toi et al. experiment on univariate outputs [Lines 73 and 74 explictly define univariate targets in their code (link)](https://github.com/ewongtoi/deep-heteroskedastic-regression/blob/master/heteroskedastic_nns/datasets/load_uci/uci_loader.py).
>
> **W3**: We describe our implementation in Appendix B. Specifically, we use separate networks to predict and regularize the mean and the covariance. While in general we do not use warm-up, we conducted experiments on the UCI regression with warm-up. This involved training the mean estimator for half the number of total training epochs, and jointly training the mean and covariance estimator for the other half. We report the results in appendix/Fig. 12. We would be happy to answer any specific queries regarding the implementation.

---

> ### Author Response · Authors · 2024-11-20
> **Author Response To Reviewer VkG5: Part 2/2**
>
> **Q1**: The major difference is in the philoshophy; since the negative log-likelihood cannot supervise the variance, Skafte et al. constructs the neighborhood on the fly to encode the variance in the batch. We avoid this by developing tools to directly supervise the variance, and pre-compute the covariance matrix to train the network for each sample directly. This also leads to better allocation of the mini-batch capacity, since the samples in the batch need not be restricted to similar neighborhoods, and therefore the mini-batch can optimize on diverse regions of the input space at the same time.
>
> Algorthmically, in Skafte et al., the neighborhood of a sample is computed based on the euclidean distance, and a threshold 'd', which is a hyperparameter. The threshold 'd' may require hyperparameter search. We make a small change: we compute neighborhoods based on the Mahalanobis distance, with number of neighbors 'k'. We use the Mahalobis distance to account for the spread and scale of the samples. We set the number of neighbors 'k' to a constant value (10 * dimensionality of output) and do not perform hyperparameter search. This is based on a heuristic that the number of samples to compute the covariance scales linearly with the dimensionality. Skafte et al. is also sensitive to the number of neighbors 'n' chosen to construct the mini-batch. Finally, in Skafte et al., the samples in the batch are no longer *i.i.d*, a common assumption in many optimization frameworks.
>
> **Q2**: While both the covariance pseudo labels and kernel methods rely on measures of similarity for the neighborhoods, we believe that they are fundamentally different. Kernel methods are used for non-linear transformations to solve challenging problems in high dimensional spaces. In contrast, we use the neighborhoods purely to compute pseudo labels for the covariance.
>
> **Q3**: While we do not impose any specific restriction on the architecture, we recommend following guidelines set by Nix and Weigend, Stirn et al. and Sluijterman et al. to decouple the mean and covariance estimators. While the two estimators can share the backbone (such as in our Human Pose experiments using Vision Transformers), the two should not strongly regularize each other.
>
> **Q4**: The biggest challenge is a mechanism to supervise the learning of the pseudo-labels. Such a mechanism is inherently infeasible with the negative log-likelihood. Therefore, it is non-trivial for methods relying on NLL to use pseudo labels.
>
> **Q5**: Since the mean and covariance estimators are decoupled, we do not employ warm-up. Therefore, the mean and covariance are learnt from the start. "Is any information for the mean relevant in the presence of pseudo labels?" We are not quite sure we understand this. We obtain pseudo labels only for the covariance, the 2-Wasserstein distance trains the mean estimator using the mean square error. Therefore the mean is also required along with the pseudo labels for the covariance.
>
> **Q6**: The pseudo labels for the covariance are computed based on the sample's label $y$. Therefore, if the mean estimator has not converged (underfit), the mean estimate is incoherent with the covariance estimate. If the mean estimator has converged, the predicted covariance will be coherent with the predicted mean.
>
> **Q7**: If the mean estimator has underfit, the predicted covariance is incoherent with the mean. However, this is unlikely in practice unless the mean estimator is strongly regularized. If it has overfit, the situation is ambiguous since it is difficult to quantify the degree of overfitting. However, we believe that the covariance predictions by themselves can still be useful. Our reasoning is based on the way the pseudo labels are being computed, which is independent of the mean estimate, and leverages patterns within the dataset. The algorithm employs the Mahalanobis distance and softmax which are differentiable, implying continuity in the pseudo labels when the labels are continuous. Therefore, the covariance estimator can learn these patterns without relying on the mean estimator, making it useful as a standalone predictor.

---

> > ### Comment · Reviewer_VkG5 · 2024-11-25
> >
> > Thank you for your replies!
> >
> > Per W3 I think I understand the training setup now. Maybe a pseudocode algorithm (in addition to the one on how the pseudo-labels are constructed) would be helpful. My main confusion was over when the pseudo-labels are incorporated to the training loop.
> >
> > For Q5 I believe what I meant to ask was if any information from the mean was relevant to the estimation of the covariance and how that would interplay with the pseudo-labels and this was answered.
> >
> > Minor: I noticed that both "pseudo label" and "pseudo-label" are used in different places throughout the paper.
> >
> > I am comfortable increasing my score.

---

> > > ### Author Response · Authors · 2024-11-25
> > >
> > > We choose the term 'pseudo-label' in the new draft, and have updated the Training paragraph in Appendix B, Experiment Details. Thank you once again for your review!

---

### Author Response · Authors · 2024-11-20
**Revised Draft: Changes**

We once again thank all the reviewers for their comments to improve the paper. We have made the following changes across the paper:

- Abstract: We explicitly mention that our analysis is for the normal distribution (*Reviewer 5LBD*).
- Deep Heteroscedastic Regression: We have updated the related works (*Reviewer VkG5*). We add a new paragraph titled preliminaries which defines the problem setting and notation (*Reviewer ZXre*).
- Analysis: We have improved the framing and formulation for Lemma 1 (*Reviewer 6hor*). Figure 2 is in log-scale for easy visualization (*Reviewer 5LBD*)
- Experiments: We move the table: UCI Regression - NLL metric from the appendix to the main paper. We report results with standardization instead of the normalization as done in previous work. Our findings remain unchanged. (*Reviewer 6hor*)
- Appendix: We add a detailed proof for Lemma 1. We also add Figure 7 to better explain the impact of residuals (*Reviewer 5LBD*). We add more experiment details (*Reviewer VkG5*). We report the mean and standard deviation in Table 3 (*Reviewer 5LBD*). We add our study of warm-up on the UCI regression datasets (*Reviewer VkG5*).

---

### Meta-Review · Area_Chair_bqfg · 2024-12-21

**Metareview:**

In the paper, the authors study self-supervised covariance estimation in deep heteroscedastic regression, which models the mean and covariance of the target distribution through neural networks. This problem is challenging due to overfitting, which can cause (co)variance estimates to collapse to zero. This paper introduces a novel approach using pseudo-labels to learn covariance. The authors evaluate fitting covariance using KL divergence and 2-Wasserstein distance, with pseudo-labels derived from a local heuristic algorithm, providing a fresh perspective on training stability and covariance estimation.

Most of the reviewers agree that the ideas and findings of the paper are interesting and novel. The empirical results are quite extensive and support the findings of the paper quite convincingly. The presentation of the paper is clear and easy to follow. After the rebuttal, most of the concerns of the reviewers were addressed.

While there are still some (minor) concerns about the comparison with previous works, I believe the current contribution and findings of the paper are sufficient. As a consequence, I recommend accepting the paper.

The authors are encouraged to incorporate the suggestions and feedbacks of the reviewers into the revision of their paper.

**Additional Comments On Reviewer Discussion:**

Please refer to the metareview.

---

### Decision · Program_Chairs · 2025-01-22

Accept (Poster)